# Transforming Growth Factor Beta 2 (TGFB2) and Interferon Gamma Receptor 2 (IFNGR2) mRNA Levels in the Brainstem Tumor Microenvironment (TME) Significantly Impact Overall Survival in Pediatric DMG Patients

**DOI:** 10.3390/biomedicines12010191

**Published:** 2024-01-15

**Authors:** Sanjive Qazi, Zahra Talebi, Vuong Trieu

**Affiliations:** Oncotelic Therapeutics, 29397 Agoura Road, Suite 107, Agoura Hills, CA 91301, USA; zahra.talebi@sapubio.com (Z.T.); vtrieu3@autotelicinc.com (V.T.)

**Keywords:** DIPG, brainstem DMG, glioma, TGF beta, TGFB2, interferon-gamma receptor, IFNGR2, JAK1, STAT1, RNAseq, mRNA

## Abstract

This hypothesis-generating study characterized the mRNA expression profiles and prognostic impacts of antigen-presenting cell (APC) markers (CD14, CD163, CD86, and ITGAX/CD11c) in pediatric brainstem diffuse midline glioma (pbDMG) tumors. We also assessed the mRNA levels of two therapeutic targets, transforming growth factor beta 2 (TGFB2) and interferon gamma receptor 2 (IFNGR2), for their biomarker potentials in these highly aggressive pbDMG tumors. The expressions of CD14, CD163, and ITGAX/CD11c mRNAs exhibited significant decreases of 1.64-fold (*p* = 0.037), 1.75-fold (*p* = 0.019), and 3.33-fold (*p* < 0.0001), respectively, in pbDMG tumors relative to those in normal brainstem/pons samples. The pbDMG samples with high levels of TGFB2 in combination with low levels of APC markers, reflecting the cold immune state of pbDMG tumors, exhibited significantly worse overall survival outcomes at low expression levels of CD14, CD163, and CD86. The expression levels of IFNGR2 and TGFB2 (1.51-fold increase (*p* = 0.002) and 1.58-fold increase (*p* = 5.5 × 10^−4^), respectively) were significantly upregulated in pbDMG tumors compared with normal brainstem/pons samples. We performed multivariate Cox proportional hazards modelling that showed TGFB2 was a prognostic indicator (HR for patients in the TGFB2^high^ group of pbDMG patients = 2.88 (1.12–7.39); *p* = 0.028) for poor overall survival (OS) and was independent of IFNGR2 levels, the age of the patient, and the significant interaction effect observed between IFNGR2 and TGFB2 (*p* = 0.015). Worse survival outcomes in pbDMG patients when comparing high versus low TGFB2 levels in the context of low IFNGR2 levels suggest that the abrogation of the TGFB2 mRNA expression in the immunologically cold tumor microenvironment can be used to treat pbDMG patients. Furthermore, pbDMG patients with low levels of JAK1 or STAT1 mRNA expression in combination with high levels of TGFB2 also exhibited poor OS outcomes, suggesting that the inclusion of (interferon-gamma) IFN-γ to stimulate and activate JAK1 and STAT1 in anti-tumor APC cells present the brainstem TME can enhance the effect of the TGFB2 blockade.

## 1. Introduction

### Diffuse Midline Gliomas (DMGs): Definition, Classification, and Treatment Options

DMGs are malignant primary glial tumors that occur in all age groups but are more prevalent in children and are responsible for 10–20% of all the central nervous system tumors and almost one-third of the high-grade gliomas in pediatric patients [1,2]. Roughly 300 pediatric cases are reported each year, with a median age at diagnosis of less than ten years [2,3].

Our understanding of DMGs’ genetic and epigenetic traits has significantly grown owing to improved biopsy techniques. A pivotal advancement in recent years has involved epigenetic investigations to identify H3K27 alterations common in DMGs and similar midline gliomas within the thalamus, spine, and brainstem regions [4,5,6,7]. These alterations can include mutations in H3.3, H3.1, or H3.2; EZHIP overexpression; or EGFR mutants [8,9,10]. According to the 2021 World Health Organization (WHO) Classification of Tumors of the Central Nervous System (CNS), pediatric-type diffuse high-grade gliomas are now classified into four types of tumors: diffuse midline gliomas, H3 K27 altered; diffuse hemispheric gliomas, H3 G34 mutant; diffuse pediatric-type high-grade gliomas, H3 wildtype and IDH (isocitrate dehydrogenase) wildtype; and infant-type hemispheric gliomas, each having distinct prognoses and molecular characteristics [10]. The pediatric brainstem DMG (formerly DIPG), the subject of our present study, is primarily diagnosed based on clinical symptoms and characteristic imaging features, and the H3K27 alternation is present in 70–85% of these DMG cases [10,11,12]. 

As for the treatment, surgical resection is usually not possible because of the tumor’s anatomic location in the pons adjacent to cranial nerve nuclei involved with autonomic functions essential for life. The standard of care management for DMGs involves fractionated radiotherapy over a six-week course, with a total dose of 54 Gy. Numerous clinical trials have evaluated cytotoxic chemotherapeutic drugs, including temozolomide, gemcitabine, and capecitabine; targeted agents, such as tyrosine kinase inhibitors; and stem cell transplantation, but none have shown improved overall survival rates [1]. An overview of current strategies for treating DMGs in pediatric patients by targeting the immune microenvironment can be found in Appendix A. These therapies mainly include adoptive immunotherapy toxicities [1,13,14], vaccines [15], oncolytic virus therapy [16,17,18,19], and immune checkpoint inhibitors [20]. Nonetheless, using immune checkpoint inhibitors as a monotherapy may not be a feasible strategy because it has been demonstrated that the expression of immune checkpoints is not overexpressed in DMG tumors. Therefore, combining ICI therapies with other methods will need to be employed to improve their therapeutic effects.

We focused our studies by targeting molecules in the pediatric brainstem DMG (pbDMG) tumor microenvironment (TME), especially because TGFB ligands play a pivotal role in generating tumors by promoting immune suppression, immune evasion, and tumor progression, which involve inhibiting CD8-antigen-positive cytotoxic T-cells and natural killer cells and activating regulatory T-cells [21,22,23,24,25,26,27,28]. Our hypothesis proposes that monocyte-derived M2 macrophages produce angiogenic and anti-inflammatory factors, such as IL-10 and TGFB ligands, to promote regulatory T-cell infiltration in the TME, leading to the downregulation of the immune response to the tumor [29,30,31]. pbDMG has a complex immune microenvironment populated by macrophages, T-cells, and B-cells [32,33]. However, the pbDMG TME environment is highly immunosuppressive owing to the prevalence of regulatory T-cells and myeloid-derived suppressor cells, which hinder a robust anti-tumor immune response. Furthermore, microglia/macrophages have been shown to express CD68 and CD163, indicating the presence of immunosuppressive macrophages [33]. Tumor-associated macrophages (TAMs) are significant in the pathophysiology of pbDMG. Early clinical research has shown anti-tumor effects in vitro using TAM-targeted interventions [34]. We suggest that inhibiting TGFB2 production and release from immunosuppression in pbDMG tumors with the concurrent activation of M0-derived anti-tumor CD86-expressing M1 macrophages via interferon-gamma (IFN-γ) may promote proinflammatory cytokine synthesis, enhanced phagocytosis, and increased tumor antigen-presenting capacity to improve the anti-tumor response [35]. 

Our previous study suggested that TGFB2 was a specific biomarker for overall and progression-free survival in the brainstem of pbDMG patients [3]. We have now extended that study with an updated cohort of pbDMG patients who exhibited high levels of tumor TGFB2 mRNA in combination with low levels of mRNA coding for IFN-γ-targetable receptors, and signaling molecules showed worse survival outcomes in pbDMG patients. Furthermore, the mRNA expression of antigen-presenting markers for macrophages, microglia, monocytes, and dendritic cells (CD14, CD163, CD86, and ITGAX/CD11c, respectively) were significantly underexpressed in pbDMG tumors. Survival curves showed that patients with low levels of these cellular markers and high levels of TGFB2 mRNA exhibited worse survival rates, which suggests that cellular-level responses can be targeted by IFN-γ; these responses have the potential to enhance the anti-tumor response when immunosuppression is lifted owing to elevated TGFB2 levels [36].

## 2. Materials and Methods

### 2.1. Comparing mRNA Expression Levels in the Brain Stem/Pons of pbDMG Tumors with Those in Normal Brainstem/Pons Tissues

Antigen-presenting cell markers CD14, CD163, CD86, and ITGAX/CD11c; transforming growth factor receptor ligands TGFB1, TGFB2, and TGFB3; IFN-γ receptor and downstream signaling molecules IFNGR2, JAK1, and STAT1; and mRNA transcript expression levels from RNA-seq experiments for brain tissues (rna_tissue_hpa.tsv.zip) were acquired from the Human Protein Atlas version 23.0 web portal (https://www.proteinatlas.org/ (accessed on 17 December 2022); the search keyword was “pons”). Human tissues were anatomically dissected and analyzed using transcriptomics and mRNA samples from normal tissues extracted from frozen tissue sections. Following the sequencing, alignment, and quantification of the extracted nuclear RNA, the genes were annotated using database Ensembl version 109 [37,38]. We compiled TPM expression values from only the pons regions of the brain by filtering “Tissue Group” annotations in the accompanying description file (“rna_tissue_hpa_description.tsv.zip”). This data file included average levels of gene expression in 29 pons regions filtered using the keyword, “pons”, which retrieved values from the following regions: “anterior cochlear nucleus, ventral”; “dorsal cochlear nucleus”; “dorsal tegmental nucleus”; “dorsolateral tegmental area”; “Kolliker–Fuse nucleus”; “lateral lemniscus nuclei”; “lateral parabrachial nucleus”; “lateral vestibular nucleus”; “locus coeruleus”; “medial olivary nucleus”; “medial parabrachial nucleus”; “medial periolivary nuclei”; “motor facial nucleus”; “motor trigeminal nucleus”; “nuclei of the trapezoid body”; “paramedian reticular nucleus”; “pontine nuclei”; “pontine raphe nucleus”; “posteroventral cochlear nucleus”; “principal sensory trigeminal nucleus”; “reticular pontine nucleus, caudal”; “reticular pontine nucleus, oral”; “reticulotegmental nucleus”; “spinal trigeminal nucleus, oral”; “subcoeruleus area”; “superior olive”; “superior vestibular nucleus”; “ventral periolivary nuclei”; and “ventrolateral tegmental area, A5 NE cell group”.

We also downloaded the mRNA expression levels of these genes from the PedcBioPortal for Childhood Cancer Genomics and compared them with mRNA expression levels in pbDMG tumors, using the reported RNAseq TPM values as detailed in [3]. Briefly, data arrays for the mRNA expression values for each gene were normalized to “transcripts per million” (“TPMs”) for gene abundance values calculated using RSEM [39] alignment algorithm. (The dataset was downloaded from the PedcBioPortal, https://pedcbioportal.kidsfirstdrc.org/study/summary?id=openpbta%2Cpbta_all (accessed on 16 May 2023) and compiled using the Open Pediatric Brain Tumor Atlas (OpenPBTA) and Pediatric Brain Tumor Atlas (PBTA, provisional) consortiums [40] (the keywordsfor the search were “brainstem glioma, diffuse intrinsic pontine glioma, diffuse midline glioma grade 4, diffuse midline glioma H2K27M WHO grade 4, diffuse midline glioma WHO grade 4 H3K27M mutant, DMG H3 K27M mutant WHO grade 4, diffuse midline high-grade glioma, diffuse hemispheric glioma H3 G34 mutant, WHO grade 4, and infiltrating DIPG”)). The PedcBioPortal enables the acquisition of CSV-formatted files for the compiled clinical metadata and expression values of the filtered patient subsets for further analyses [41,42,43,44].

We utilized the TPM expression values from the 29 pons regions of the brain and filtered annotations under “tissue group” in the description file to compare with those of 45 pbDMG patients by applying a two-way ANOVA model to identify differentially expressed genes. The log2-transformed TPM values for the genes (model 1: TGFB1, TGFB2, TGFB3, JAK1, STAT1, and IFNGR2; model 2: CD14, CD163, CD86, and ITGAX/CD11c) and tissues (29 normal pons tissues; 45 brainstem/pons specimens from pbDMG patients for both models 1 and 2) were included as fixed factors, along with one interaction term to investigate gene-level effects for normal and pbDMG tissues (gene x tissue). For each gene, we conducted a comparison between normal pons and pbDMG samples and then determined the significance by adjusting the *p*-value using the false discovery rate algorithm provided in the R-package (FDR corrected for all the pairs in model 1 and blocked the design at the gene level in model 2). The calculations were performed in R using the multcomp_1.4-17 and emmeans_1.7.0 packages run in R, version 4.1.2, with the RStudio front end (RStudio 2021.09.0+351 “Ghost Orchid” Release). Bar chart graphics were constructed using the ggplot2_3.3.5 R package.

### 2.2. Comparison of Characteristics for pbDMG Patients with Low-Grade Gliomas and High-Grade Gliomas and Stratification of Patient Subsets Relative to mRNA Expression Levels

The clinical metadata and RNA sequencing data for 45 pbDMG patients were analyzed to stratify patients according to mRNA expression levels in this study. The diagnosis of pbDMG was primarily ascertained using radiological methods, which revealed borderless, diffuse, expansile, hyperintense lesions in the pons, which extended to other areas of the brainstem for all 45 patients. (Forty-four of these patients had specified lesions from the pons, and one patient harbored the DMG/H3K27M mutation in the brainstem/medulla region). Upon examining the clinical metadata file, it was reported that 31 out of the 45 patients had diffuse midline gliomas with the H3K27M mutation (DMG/H3K27M; “diffuse midline glioma H3 K27 altered” according to the WHO 2021 classification scheme, which designates these tumors as Grade 4). Thirteen tumors were designated as WHO Grade 3/4 high-grade gliomas (HGGs), and one tumor had no designation (NA) [44]. Two HGG patients were reported to harbor both H3 and IDH wildtype genes, and one patient harbored H3 wildtype and IDH wildtype genes and a TP53 mutation. Assays for mRNA expression values were obtained from 22 deceased patients: 1 at diagnosis, 20 from the initial CNS tumors, and 2 from progressive disease patients. The patient characteristics for these 45 pbDMG patients were compared using 171 pediatric high-grade gliomas and 404 low-grade gliomas obtained from the PBTA database (downloaded from the PedcBioPortal, https://pedcbioportal.kidsfirstdrc.org/study/summary?id=openpbta%2Cpbta_all (accessed on 16 May 2023), and compiled using the Open Pediatric Brain Tumor Atlas (OpenPBTA) and Pediatric Brain Tumor Atlas (PBTA, provisional) consortiums [40] (the keywordsfor the high-grade glioma and low-grade glioma searches were “CANCER_TYPE_DETAILED: low-grade glioma, NOS, or high-grade glioma, NOS RNA expression” with “ONCOTREE_CODE: DIPG, hggnos, and lggnos).

The distribution of the age (median = 7 years; (range = 2–18 years)), fraction of the altered genome (median = 0.16; (range = 0–0.81)), and mutation count (median = 23; (range = 2–499)) for the 45 pbDMG patients were within the distribution observed for the HGG and LGG patients (Appendix A). We examined the effect of the gene-level mRNA expression on OS outcomes by compiling patient-level data. The current regimens for treating pbDMG patients via https://clinicaltrials.gov/ct2/show/NCT02274987) (accessed on 1 January 2023) have established strategies involving the use of standard radiation therapy followed by specialized therapy with targeted FDA-approved drugs. The treatment is guided by gene expression analysis, whole-exome sequencing, and biomarkers [45,46]. The TPM metric was used to calculate the percentiles of CD14, CD163, CD86, ITGAX/CD11c, TGFB1/2/3, JAK1, STAT1, and IFNGR2 expression in 45 pbDMG patients. Four patient groups were then formed based on their expression levels of TGFB2 and IFNGR2: high expressions of both (TGFB2^high^/IFNGR2^high^; higher than or equal to those of both TGFB2 and IFNGR2 in the 50th percentile); low expressions of both (TGFB2^low^/IFNGR2^low^; lower than those of both TGFB2 and IFNGR2 in the 50th percentile); and combinations of high and low expression levels for both (TGFB2^high^/IFNGR2^low^ and TGFB2^low^/IFNGR2^high^).

### 2.3. Overall Survival (OS) Outcomes of pbDMG Patients Stratified Relative to TGFB2 and IFNGR2/JAK1/STAT1 mRNA Expression Levels

OS curves were then compared between these groups to assess the survival impacts of the combinations of the four stratified groups of TGFB2 and IFNGR2 levels. We also compared the impacts of TGFB2^high^/IFNGR2^low^, TGFB2^high^/JAK1^low^, and TGFB2^high^/STAT1^low^ versus the remaining patients on the OS to test the effect of the IFNGR2/JAK1/STAT1 axis on the survival of these patients. To test for the specificity of the impact of TGFB2, we also compared TGFB1^high^/IFNGR2^low^ versus those of the remaining patients and TGFB3^high^/IFNGR2^low^ versus those of the remaining patients for OS outcomes. Similarly, we assessed the impacts of high levels of TGFB2 expression in combination with low levels of expression of the four antigen-presenting cell surface makers, CD14, CD163, CD86, and ITGAX/CD11c, compared with those of the remaining patients. These comparisons of OS outcomes in the patient subsets were carried out using the Kaplan–Meier (KM) method, and the statistical significance was tested using the log-rank chi-square test and the following R-based software packages: survival_3.2-13, survminer_0.4.9, and survMisc_0.5.5. Graphical representations of the treatment outcomes were visualized using the following graph-drawing packages implemented in R: dplyr_1.0.7, ggplot2_3.3.5, and ggthemes_4.2.4. We considered *p*-values less than 0.05 significant after correcting for multiple comparisons across the four groups (6 comparisons), using the Benjamini–Hochberg method.

### 2.4. Multivariate Analysis of OS Outcomes for pbDMG Patients Stratified Relative to TGFB2 and IFNGR2 mRNA Expression Levels and Controlled for Age and Interaction of TGFB2 and IFNGR2

To determine the potentially independent impacts of the TGFB2 and IFNGR2 levels on the OS, multivariate analyses were conducted using the Cox proportional hazards model, whereby age and the interaction between TGFB2 and IFNGR2 were controlled for in the analysis. Briefly, the model included (i) the mRNA expression level for TGFB2 as a categorical variable comparing high versus low TGFB2 mRNA expression levels at a 50% cutoff for expression values; (ii) the mRNA expression level for IFNGR2 as a categorical variable comparing high versus low IFNGR2 mRNA expression levels (50% cutoff), and (iii) the age implemented in R (survival_3.2-13 run in R, version 4.1.2.). Forest plots were utilized to visualize the hazard ratios for the Cox proportional hazards models for OS outcomes (survminer_0.4.9 run in R, version 4.1.2 (1 November 2021)). The life table hazard ratios (HRs) were estimated using the exponentiated regression coefficient for Cox proportional hazards analyses implemented in R (survival_3.2-13 run in R, version 4.1.2). We also investigated the impact of including an interaction term as the fourth parameter in the Cox proportional hazards model (TGFB2 x IFNGR2) to compare the independent effects of TGFB2 and IFNGR2 in models with and without the interaction term. To visualize the predicted survival proportion at any given time for combinations of high and low TGFB2 mRNA expression groups in the context of high and low IFNGR2 mRNA expression groups from the interaction model, we plotted and calculated the shift in the baseline OS curve for the 45 pbDMG patients from the fitted hazard functions. In these comparisons, we compared the median OS times for the TGFB2^high^ versus TGFB2^low^ groups of patients in patients who expressed either high or low levels of IFNGR2. A significant interaction effect from the interaction model would indicate differences in OS times for the TGFB2^high^ versus TGFB2^low^ groups of patients, depending on the level of IFNGR2.

## 3. Results

### 3.1. Downregulation of Markers for Anti-Tumor Antigen-Presenting Cells (APCs) in pbDMG Tumors

The expressions of CD14, CD163, and ITGAX mRNAs exhibited significant decreases of 1.64-fold (*p* = 0.037), 1.75-fold (*p* = 0.019), and 3.33-fold (*p* < 0.0001), respectively, in pbDMG tumors. The CD86 mRNA expression in pbDMG patients showed a non-significant 1.42-fold decrease compared with that in normal brainstem/pons tissue (*p* = 0.14) (Appendix A). We investigated the prognostic impact of high levels of TGFB2 in combination with low levels of these markers for antigen-presenting cells in pbDMG patients (Appendix A). Patients with low levels of CD14 and CD163 expression in combination with high levels of TGFB2 exhibited worse OS outcomes, and these makers were also expressed at lower levels in pbDMG tumors compared with those in normal brainstem/pons tissue (Appendix A). The median overall survival time for 14 patients in the TGFB2^high^/CD14^low^ group was 7.5 months (95% CI: 5–12; number of events = 14), which was significantly shorter (log-rank *p*-value = 0.007) than the median survival time for the 31 remaining patients at 13 months (95% CI: 8–15; number of events = 29) (Appendix A), whereas the median OS time for 9 patients in the TGFB2^high^/CD163^low^ group was 7 months (95% CI: 5–NA; number of events = 9) as compared with the median OS time for the 36 remaining patients at 11 months (95% CI: 8–15; number of events = 34; log-rank *p*-value = 0.014) (Appendix A). The median OS time for 13 patients in the TGFB2^high^/CD86^low^ group was 7 months (95% CI: 5–NA; number of events = 13), which was significantly shorter than the overall survival time for the 32 remaining patients at 13 months (95% CI: 8–15; number of events = 30) (log-rank *p*-value = 0.001) (Appendix A), and the median OS time for 9 patients in the TGFB2^high^/ITGAX^low^ group was 7 months (95% CI: 5–NA; number of events = 9), which was shorter than but not statistically significant compared with the median overall survival time for the remaining 36 patients at 11 months (95% CI: 8–14; number of events = 34) (*p* = 0.885) (Appendix A).

### 3.2. Amplified Expression of TGFB2 Compared with Those of TGFB1 and TGFB3 mRNAs in pbDMG Patients and Normal Pons Tissue

The brainstem/pons tissues from pbDMG patients showed a selective upregulation of TGFB2 and downregulations of TGFB1 and TGFB3 when compared with those of the normal pons tissue, with a significant 2.84-fold decrease, 1.51-fold increase, and 4.08-fold decrease in TGFB1, TGFB2, and TGFB3 mRNA expressions (*p* < 0.0001, 0.002, and <0.0001, respectively) (Appendix A). The profile of the TGFB ligands differed markedly in the pbDMG brain stem/pons and normal pons tissues, whereby in the pbDMG samples, the TGFB2 mRNA expression was significantly higher than those of the TGFB1 (1.54-fold increase; *p* = 4.1 × 10^−4^) and TGFB3 (2.25-fold increase; *p* < 0.0001), suggesting the specific upregulation of the TGFB2 isoform in the pbDMG tumor tissue. When comparing the mRNAs of TGFB1 and TGFB2 in the normal pons tissue, it was found that TGFB2 had a significantly lower expression, with a 2.78-fold decrease (*p* < 0.0001). Similarly, when comparing the mRNA expressions of TGFB2 and TGFB3, TGFB2 showed a highly significant 2.74-fold decrease in mRNA levels (*p* < 0.0001) (Appendix A).

### 3.3. Amplified Expression of IFNGR2 mRNA in the Brainstem/Pons Region of pbDMG Patients Compared with That in Normal Pons Tissue

In contrast to the significant downregulations of CD14, CD163, and ITGAX mRNAs in pbDMG tumors, we observed a significant upregulation of IFNGR2 mRNA levels compared with those in normal pons tissue (1.58-fold increase; *p* = 5.5 × 10^−4^). Taken together, the expression profiles for the TGFB2 ligand and IFNGR2 mRNA expression point to an important role as prognostic indicators in the progression of pbDMG tumors (Figure 1). The expression of JAK1 was similar in both pbDMG tumors and normal brainstem/pons tissue (*p* = 0.73) and exhibited an order of magnitude similar to that of IFNGR2 expression in pbDMG tumors (Figure 1). Examination of the STAT1 expression showed significantly lower levels of expression (*p* = 0.0029) in pbDMG tumors compared with those in normal brainstem/pons tissue (Figure 1).

### 3.4. Amplified TGFB2 Expression in Combination with Reduced IFNGR2 Expression in pbDMG Patients Is Associated with Worse OS Outcomes

Our investigation focused on assessing the impacts of the TGFB2 and IFNGR2 expression levels on the overall survival of four groups of pbDMG patients and compared low levels of mRNA expression for both TGFB2 and IFNGR2 (TGFB2^low^/IFNGR2^low^; lower than those of both TGFB2 and IFNGR2 in the 50th percentile), combinations of high and low expression levels for TGFB2 and IFNGR2 (TGFB2^low^/IFNGR2^high^ and TGFB2^high^/IFNGR2^low^), and high expressions of both TGFB2 and IFNGR2 (TGFB2^high^/IFNGR2^high^; higher than or equal to those of both TGFB2 and IFNGR2 the 50th percentile). Examination of pairwise differences for median OS times showed significant differences for the comparison between the TGFB2^low^/IFNGR2^high^ (median = 13 months (95% CI = 10–NA months)) and TGFB2^high^/IFNGR2^low^ groups of patients (median = 7 months (95% CI = 5–NA months); *p* = 0.009) and for the comparison between the TGFB2^high^/IFNGR2^low^ and TGFB2^high^/IFNGR2^high^ (median = 15 months (95% CI = 7–NA months)) groups of patients (*p* = 0.012) (Figure 2). These results suggest that high levels of IFNGR2 confer a significant survival benefit at high TGFB2 mRNA expression levels. At low levels of IFNGR2 expression, pbDMG patients with high levels of TGFB2 exhibit worse survival outcomes than the remaining patients (Figure 3). Out of the 45 patients who were studied, a group of 13 had high levels of TGFB2 and low levels of IFNGR2. Their median overall survival time was 7 months, which significantly contrasted with that of the remaining group of 32 patients, with a median overall survival time of 13 months (95% CI = 10–15 months). The survival outcome showed a significant difference between the two groups, with a log-rank chi-square value of 13.5 and a *p*-value of 2.3 × 10^−4^, thereby suggesting a significant prognostic impact on pbDMG patients expressing high levels of TGFB2 and low levels of IFNGR2 (Figure 3). However, pbDMG patients with low levels of IFNGR2 and high levels of either TGFB1 (Appendix A) or TGFB3 (Appendix A) did not exhibit worse OS times, suggesting that the prognostic impact of TGFB2 in the context of low IFNGR2 expression was specific to TGFB2.

### 3.5. Amplified TGFB2 Levels Are an Independent Negative Prognostic Indicator for OS When Controlling for Age and IFNGR2 Levels

We further investigated the effects of TGFB2 and IFNGR2 levels in a multivariate context, including age as a linear control variable, utilizing Cox proportional hazards models (Figure 4A). This model, without an interaction term, showed that there was no significant increase in HR for the TGFB2^high^ group of patients (HR (95% CI range) = 1.29 (0.66–2.49); *p* = 0.457). However, there was a significant decrease in HR for the IFNGR2^high^ group of patients (HR (95% CI range) = 0.38 (0.19–0.75); *p* = 0.006), suggesting that high IFNGR2 levels have an overall pro-survival benefit. The inclusion of an interaction term in the Cox proportional hazards model (Figure 4B) generated a more complex effect that uncovered a significant increase in the hazard ratio for patients in the TGFB2^high^ group (HR (95% CI range) = 2.88 (1.12–7.39); *p* = 0.028), which was independent of IFNGR2 levels and age, and the significant pro-survival effect of high levels of IFNGR2 observed in the model without the interaction term was now captured in the significant effect of the IFNGR2 x TGFB2 interaction term ((HR (95% CI range) = 0.17 (0.04–0.72); *p* = 0.015). The parameters were analyzed in the Cox proportional hazards regression model, which factored in the interaction term for combinations of high and low TGFB2 mRNA expression groups within the context of low levels and high levels of IFNGR2 mRNA expression in groups of patients. This suggests that when the IFNGR2 expression is low, higher levels of TGFB2 lead to worse overall survival rates. Patients in the TGFB2^low^ group had a median OS time of 11 months, which was higher than the upper 95% confidence limit for the TGFB2^high^ group (upper 95% confidence interval = 10 months). However, when the IFNGR2 expression is high, the TGFB2^high^ group had a median OS time of 15 months, which falls within the 95% confidence interval for the TGFB2^low^ group (median = 13 months; 95% CI = 10–17 months) (Figure 5).

### 3.6. Augmented TGFB2 mRNA Levels and Reduced Levels of Signaling Molecules Downstream of Interferon Receptor Activation Are Significant Negative Prognostic Indicators for OS in pbDMG Patients

We next investigated the prognostic impacts of JAK1 and STAT1 mRNA expression levels, which are downstream signaling molecules of IFNGR2 activation, in combination with TGFB2 to assess the biochemical functional significance of the interaction observed between TGFB2 and IFNGR2. OS curves were then compared between two groups of patients, TGFB2^high^/JAK1^low^ versus the remaining patients, to assess the survival impact of the combination of the TGFB2 and JAK1 levels. Out of the 45 patients who were studied, the 11 patients in the TGFB2^high^/JAK1^low^ group had a median overall survival time of 5 months (95% CI: 4–NA; number of events = 11), while the remaining 34 patients had a median overall survival time of 13 months (95% CI: 9–15; number of events = 32). This difference in survival was found to be statistically significant (log-rank chi-square value = 13.5; *p*-value = 2.4 × 10^−4^) (Figure 6). Similarly, the 11 patients in the TGFB2^high^/STAT1^low^ group had a median overall survival time of 7 months (95% CI: 4–NA; number of events = 11), while the remaining 34 patients had a median overall survival time of 13 months (95% CI: 8–15; number of events = 32). The difference in survival between these groups was also found to be statistically significant (log-rank chi-square value = 10.3; *p*-value = 0.0014) (Figure 7).

## 4. Discussion

We investigated the mRNA expression profiles of markers of antigen-presenting cells in tumors isolated from pbDMG patients. In this analysis, the expressions of CD14, CD163, and ITGAX mRNAs exhibited significant decreases of 1.64-fold (*p* = 0.037), 1.75-fold (*p* = 0.019), and 3.33-fold (*p* < 0.0001), respectively, in pbDMG tumors relative to normal brainstem/pons samples. CD86 mRNA expression in pbDMG patients showed a non-significant 1.42-fold decrease compared with those in normal brainstem/pons tissue (*p* = 0.14) (Appendix A). In these pbDMG patients, high levels of TGFB2 expression in combination with low levels of CD14 (Appendix A), CD163 (Appendix A), and CD86 (Appendix A) expressions exhibited significantly worse OS outcomes than in the remaining patients, suggesting that abrogating TGFB2 and increasing the infiltration and/or function of antigen-presenting cells has the potential to markedly improve the prognosis of pbDMG patients. Taken together, these analysis results suggest that reduced expressions of CD14 and CD163 in pbDMG tumors with high TGFB2 and low CD14 or CD163 expressions result in poor OS outcomes. CD14 presents as a marker for monocytes, dendritic cells, macrophages, and microglia in tumors [30,47,48,49,50,51,52], whereas CD163 expression serves as a marker for M2-like TAMs [30,50,51] and microglia [53]. Microglia and macrophages make up nearly half of all the glioblastoma tumors. Both cell types express CD14 or CD163, but the microglial composition can be quantified using the specific microglial markers P2RY12 and TMEM119 [53]. An increase in the microglia-to-TAM ratio has been shown to correlate with better survival outcomes in patients with glioblastoma [53]. Therefore, increased CD14 or CD163 expression in microglia may have contributed to longer OS times in our study of pbDMGs. Interestingly, in colorectal cancers (CRCs), CD163-positive M2-like macrophages exhibit anti-tumor activity in tumors with low levels of the anti-phagocytotic marker CD47 and positively correlate with the expression of CD68, which predicts increases in long-term survival [49]. The increased density of CD163-positive TAMs, along with the high CD68 expression, was also associated with upregulated immune signaling and improved outcomes for gastric cancer [48]. CD86 is a marker for proinflammatory M1-like TAMs [29,31,51] and leads to worse survival outcomes at low expression levels in combination with high TGFB2 levels in pbDMG tumors, suggesting that stimulating the expression of CD86 and abrogation of TGFB2 would result in more favorable OS outcomes. The expression of ITGAX/CD11c was significantly reduced in pbDMG tumors but had no prognostic impact on OS times, suggesting the reduced presence of antigen-presenting dendritic cells in the pbDMG tumors.

Our present study shows that high levels of TGFB2 expression in combination with low levels of markers of antigen-presenting cells resulted in worse OS outcomes in 45 pbDMG patients (Appendix A), as extended from our previously conducted study assessing the importance of increased levels of transforming growth factor TGFB2 in tumor tissues of 41 children with pbDMGs [3]. High intra-tumor TGFB2 levels in the upper quartile range of the mRNA expression in the 41 pbDMG patients were associated with poor treatment and overall survival rates. (The mean log2-TPM expression level for TGFB2 in the 11-patient TGFB2^high^ subset was 6.19 ± 0.24 (median; range = 5.81; 5.23–7.63, respectively). The median overall survival for the 11-patient TGFB2 subset was 5 months, while the median overall survival for the 30-patient TGFB2^low^ subset was 11.5 months in this group of pbDMG patients (*p* < 0.001) [3]. These present findings also suggested that TGFB2 is a potential therapeutic target for pbDMG treatment and extended the potential role of IFNGR2 acting in combination with TGFB2 in the TME. The current analysis was performed on an expanded 45-patient pbDMG dataset that included a more detailed molecular subtype characterization of the brainstem/pons tissue biopsied from these patients and examined the prognostic interaction of TGFB2 and IFNGR2 mRNA expressions for OS outcomes. As per our previous findings, TGFB2 levels were significantly upregulated in pbDMG samples compared with those in the normal pons (Appendix A) and exhibited higher levels of expression than TGFB1 and TGFB3 in the pbDMG samples.

In contrast to antigen-presenting cell markers CD14, CD163, and ITGAX, which exhibited downregulated expression in pbDMG patients relative to those in normal brainstem/pons tissue, the IFNGR2 mRNA expression was augmented in pbDMG patients, indicating that this receptor could be targeted for treating pbDMG patients (Figure 1). The examination of the overall survival impact of the four pbDMG patient groupings based on the expression levels of TGFB2 and IFNGR2 demonstrated a significant prognostic impact for patients with pbDMG who exhibited a high expression of TGFB2 and a low expression of IFNGR2 (Figure 2), whereby significant differences were observed for the comparison between the TGFB2^low^/IFNGR2^high^ (median = 13 months (95% CI = 10–NA months)) and TGFB2^high^/IFNGR2^low^ (median = 7 months; *p* = 0.009) groups of patients and for the comparison between the TGFB2^high^/IFNGR2^low^ and TGFB2^high^/IFNGR2^high^ (median = 15 months) groups of patients (*p* = 0.012) (Figure 2). The mean (±SEM) and median (range) log2-TPM mRNA expression values of TGFB2 in the 13-patient TGFB2^high^/IFNGR2^low^ subset were 5.3 ± 0.3 and 5.2 (4.1–7), respectively. Because the expression value range in the present study included pbDMG patients who exhibited TGFB2 expressions greater than or equal to the median value, there was a lower threshold for TGFB2 in the 13-patient subset, which resulted in worse OS outcomes when considering patients with low expressions of IFNGR2 (<median log2-TPM for IFNGR2), in comparison with the range of log2-TPM values for TGFB2 expression reported in our previous study, which only considered TGFB2 expression (range = 5.23–7.63 for 11 patients with TGFB2^high^ compared with range = 4.1–7 for 13 patients with TGFB2^high^/IFNGR2^low^). Therefore, our present study suggested that low TGFB2 expression levels exert a significant prognostic impact on patients with low levels of IFNGR2 expression.

The immune cold state of the pbDMG tumor microenvironment makes it challenging for treating these aggressive tumors [1]. Our study characterizes the prognostic impacts of TGFB2 and IFNGR2 in brainstem tumors, which provide potential insights into the prognostic role of these molecules in pbDMG patients. Several immune cells express receptors for interferon [36], and interferon-gamma (IFN-γ) is crucial in promoting a proinflammatory tumor environment and enhancing tumor immunogenicity by inducing M1 macrophages. M1 macrophages secrete IFN-γ, which helps to create a proinflammatory microenvironment and promote the development of the T-cell response [35]. Macrophages cannot become tumoricidal with IFN-γ alone because it requires combinations with toll-like receptor (TLR) agonists to induce macrophage tumoricidal activity. For the optimal development of anti-tumor M1 macrophages, two signals from the microenvironment, namely IFN-γ and TLR agonists, are thought to be necessary [54]. These present studies detail the interaction between TGFB2 and IFNGR2 in the tumor microenvironment, resulting in a prognostic impact on OS outcomes in pbDMG patients. We applied a Cox proportional hazards model that investigated the independent effect of TGFB2 mRNA levels, considering the effects of IFNGR2 levels, age, and the interaction of IFNGR2 and TGFB2 levels in pbDMG patients (Figure 4B). This model uncovered a significant increase in the hazard ratio for patients in the TGFB2^high^ group (HR (95% CI range) = 2.88 (1.12–7.39); *p* = 0.028), which was independent of the IFNGR2 levels and age. The significant interaction between TGFB2 and IFNGR2 was visualized by modeling the survival proportion, which factored in the interaction term for combinations of high and low TGFB2 mRNA expression groups within the context of low levels and high levels of IFNGR2 mRNA expression groups in patients (Figure 5). In the context of low levels of IFNGR2 expression, higher levels of TGFB2 lead to worse overall survival rates than patients expressing lower levels of TGFB2. Patients in the TGFB2^low^ group had a median OS time of 11 months, greater than those in the upper 95% confidence limit for the TGFB2^high^ group (upper 95% confidence interval = 10 months). These results suggest that the OS impact of high levels of TGFB2 mRNA expression is most prominent in the tumor microenvironment expressing low levels of IFNGR2 (Figure 5). We suggest that high levels of TGFB2 combined with low levels of IFNGR2 are strong prognostic markers for worse overall survival outcomes and that abrogating TGFB2 levels in tumors with low levels of IFNGR2 in anti-tumor immune cells is a feasible strategy to improve overall survival in pbDMG patients.

IFN-γ activates interferon-gamma receptors in various immune cells, including T lymphocytes, cytotoxic T lymphocytes (CTLs), natural killer cells (NK), natural killer T-cells (NKT), dendritic cells (DCs), B-cells, and myeloid cells [36,55]. IFN-γ binds to the IFNGR1/IFNGR2 heterodimer receptor to activate Janus kinases JAK1 and JAK2 and the signal transducer and activator of transcription STAT1 and STAT3 pathways (Figure 8) [35,56]. We examined whether high levels of TGFB2 and low levels of either JAK1 or STAT1 also relate to worse survival outcomes and, hence, suggest that the IFNGR2/TGFB2 interaction has functional significance. In this study of 45 patients, it was observed that the 11 patients in the TGFB2^high^/JAK1^low^ group had a median overall survival time of 5 months (95% CI: 4–NA; number of events = 11). In contrast, the remaining 34 patients had a median overall survival time of 13 months (95% CI: 9–15; number of events = 32; *p* = 2.4 × 10^−4^), as shown in Figure 6. Similarly, the 11 patients in the TGFB2^high^/STAT1^low^ group had a median overall survival time of 7 months (95% CI: 4–NA; number of events = 11) compared with the remaining 34 patients, who had a median overall survival time of 13 months (95% CI: 8–15; number of events = 32). The difference in the survival between these groups was also found to be statistically significant (log-rank chi-square value = 10.3; *p* = 0.0014), as depicted in Figure 7. The prognostic significance of low levels of the IFNGR2 receptor and the downstream signaling molecules JAK1 and STAT1 suggest that activation with the IFN-γ ligand can improve the overall survival outcomes in pbDMG patients.

Our study suggests that treatment options that abrogate the TGFB2 mRNA expression, such as OT-101 [57,58], combined with the delivery of interferon-gamma to the tumor microenvironment can improve the survival outcomes of pbDMG patients. IFN-γ has the potential to exert its anti-tumor effects by acting directly on malignant cells, as well as other cells in the TME, including endothelial cells and immune cells, such as CD86 positive M0-derived M1-like macrophages [30,59]. IFN-γ can be administered for treatment through various routes and delivery systems, including liposomal formulations [55,60,61]. It has been shown that delivering IFN-γ locally can reduce tumor growth by stimulating antigen presentation in the myeloid compartment and on leukemia cells, resulting in the recruitment and activation of T-cells [62]. A proof of concept for targeting the delivery of interferons was implicated by a study that showed IFN-γ gene delivery can effectively treat airway hyperresponsiveness and eosinophilia in mice through both intravenous and intratracheal delivery methods [63].

**Figure 8 biomedicines-12-00191-f008:**
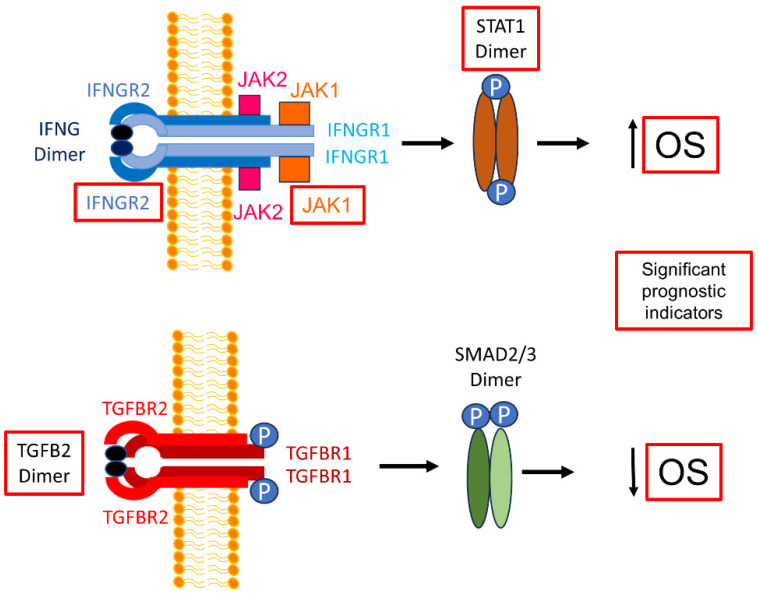
Opposite effects of increasing TGFB2 and IFNGR2/JAK1/STAT1 mRNA levels on OS outcomes in pbDMG patients. The schematic displays simplified models for the activation of the interferon-gamma receptor [56] and transforming growth factor beta receptor [64] on the plasma membrane by their respective ligands, IFN-γ (IFNG) and TGFB2, leading to a more favorable OS for IFNGR2/JAK1/STAT1 and a worse OS for TGFB2. The red boxes depict significant prognostic indicators. IFN-γ dimer signals by binding to the extracellular domains of a cell surface receptor composed of an IFNGR1 and IFNGR2 tetrameric structure (blue structure). The intracellular domains of these receptors contain motifs that bind to JAK1 (orange box), associated with IFNGR1 (light blue), and JAK2 (pink box), associated with IFNGR2 (dark blue). JAK1/2 phosphorylate(s) STAT1 monomers to form STAT1 dimers (brown ellipses attached to blue circles depicting phosphorylation), which enter the nucleus to activate transcription. TGFB2 dimer binds to a tetrameric receptor structure composed of TGFBR1 (2 dark red subunits) and TGFBR2 (2 red subunits). This facilitates the phosphorylation of TGFBR1 by TGFBR2, subsequently resulting in SMAD2/3 phosphorylation and activation by the TGFB receptor kinase. Phosphorylated SMAD2/3 enters the nucleus to activate gene transcription.

It is important to note that the current study has some limitations. First, the bioinformatics-based analyses used from archived datasets were not supplemented with additional data from the laboratory testing of TGFB2 and IFNGR2 mRNA levels, using RT-qPCR methods. Second, we could not thoroughly analyze OS outcomes correlated to specific treatments owing to the absence of patient-specific treatment information in the database. To verify the negative outcome of the TGFB2 status, a validation study based on hypothesis testing will be necessary. This study should include RT-qPCR and RNAseq data and be carried out on a larger population of pbDMG patients who have received a contemporary standard of care treatment.

## 5. Conclusions

The expressions of the APC markers CD14, CD163, and ITGAX/CD11c mRNAs were shown to be significantly lower in pbDMG tumors relative to normal brainstem/pons samples. pbDMG samples with high levels of TGFB2 mRNA in combination with low levels of APC markers reflected the cold immune state of pbDMG tumors, which correlated with significantly worse overall survival outcomes in patients with low expression levels of CD14, CD163, and CD86. IFNGR2 and TGFB2 expression levels were significantly higher in pbDMG tumors than in normal brainstem/pons samples, and high TGFB2 levels were linked to poor overall survival (OS) in pbDMG patients (Figure 8), regardless of the IFNGR2 levels or patients’ age; therefore, the abrogation of TGFB2 mRNA expression in the immunologically cold tumor microenvironment may help to treat pbDMG patients. High levels of IFNGR2, JAK1, and STAT1 correlate with an improved pbDMG OS (Figure 8). Additionally, pbDMG patients with low levels of JAK1 or STAT1 mRNA expression in combination with high levels of TGFB2 exhibited poor OS outcomes, suggesting that stimulating and activating JAK1 and STAT1 in anti-tumor APC cells, using IFN-γ in the brainstem tumor microenvironment could enhance the effect of the TGFB2 blockade.

## Figures and Tables

**Figure 1 biomedicines-12-00191-f001:**
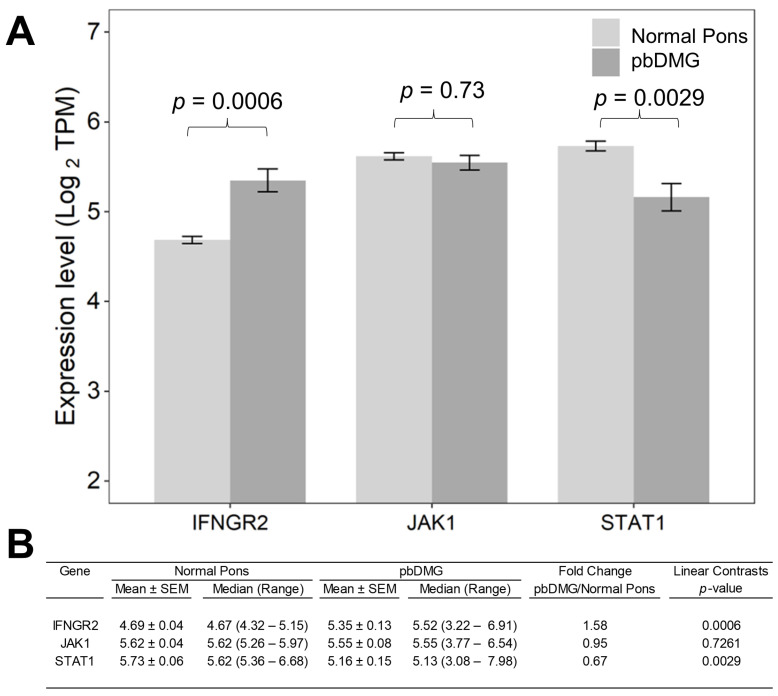
Upregulation of IFNGR2 mRNA expression in pbDMG tumor samples. We obtained IFNGR2, JAK1, and STAT1 mRNA expression levels (log2-transformed TPM) for primary tumor samples from pbDMG patients, including molecular subtype classifications of DMG, H3K27M (*n* = 23); DMG, H3K27M and TP53 (*n* = 8); HGG, H3 wildtype (*n* = 2); HGG, H3 wildtype and TP53 (*n* = 1); HGG, to be classified (*n* = 10) and 1 not determined, whose brain tumors were localized to the pons/brainstem. IFNGR2 (*n* = 45), JAK1 (*n* = 45), and STAT1 (*n* = 45) mRNA expression levels (log2 TPM) in DMG samples were compared with the expression levels in normal pons samples from 29 pons regions (downloaded from https://www.proteinatlas.org/about/download (accessed on 17 December 2022)). The expression for these 29 normal pons regions was determined by averaging the TPM values for 2–8 independent samples/regions from 21 subjects. (**A**) The bar charts illustrate mean expression levels for mRNA in tumor specimens from DMG patients (dark gray bars) compared with those in normal pons samples (light gray bars). (**B**) The statistical significance of differences in mRNA expression levels (in log2-transformed TPM values) was assessed using a two-way ANOVA with linear contrasts and FDR-adjusted *p*-values. There was a significant augmentation in the IFNGR2 mRNA levels in DMG samples compared with that in the normal pons tissue (1.58-fold increase; *p* = 5.5 × 10^−4^).

**Figure 2 biomedicines-12-00191-f002:**
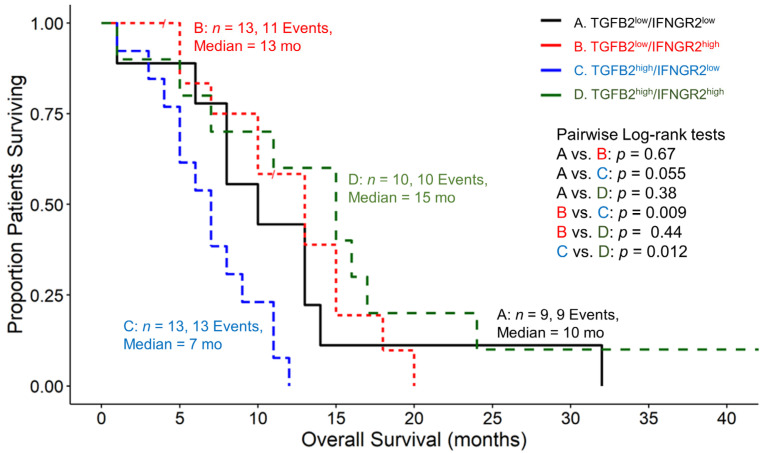
pbDMG patients with high levels of TGFB2 and low levels of IFNGR2 mRNA expression exhibited significantly shorter OS times than patients with high levels of IFNGR2 mRNA expression. We downloaded clinical metadata and RNA-sequencing-based mRNA expression data for 45 patients diagnosed with pbDMG (https://pedcbioportal.kidsfirstdrc.org/study/summary?id=openpbta%2Cpbta_all (accessed on 16 May 2023)). The RSEM-determined TPM metric was used to calculate the percentiles of TGFB2 and IFNGR2 expressions in 45 pbDMG patients. Four patient groups were then formed based on their expression levels of TGFB2 and IFNGR2: low expressions of both ((A) TGFB2^low^/IFNGR2^low^; lower than those of both TGFB2 and IFNGR2 in the 50th percentile), combinations of high and low expression levels for both (B) TGFB2^low^/IFNGR2^high^ and (C) TGFB2^high^/IFNGR2^low^, and high expression levels of both ((D) TGFB2^high^/IFNGR2^high^; higher than or equal to those of both TGFB2 and IFNGR2 in the 50th percentile). OS curves were then compared between these groups to assess the survival impacts of the combinations of TGFB2 and IFNGR2 levels. The patient groups’ survival times were recorded as follows: TGFB2^low^/IFNGR2^low^ had a median survival time of 10 months (with 95% CI ranging from 8 to NA and 9 events); TGFB2^low^/IFNGR2^high^ had a median survival time of 13 months (with 95% CI ranging from 10 to NA and 11 events); TGFB2^high^/IFNGR2^low^ had a median survival time of 7 months (with 95% CI ranging from 5 to NA and 13 events); and TGFB2^high^/IFNGR2^high^ had a median survival time of 15 months (with 95% CI ranging from 7 to NA and 10 events). Examination of pairwise log-rank differences in OS times showed significant differences for the comparison between the TGFB2^low^/IFNGR2^high^ and TGFB2^high^/IFNGR2^low^ groups of patients (*p* = 0.009) and for the comparison between the TGFB2^high^/IFNGR2^low^ and TGFB2^high^/IFNGR2^high^ groups of patients (*p* = 0.012).

**Figure 3 biomedicines-12-00191-f003:**
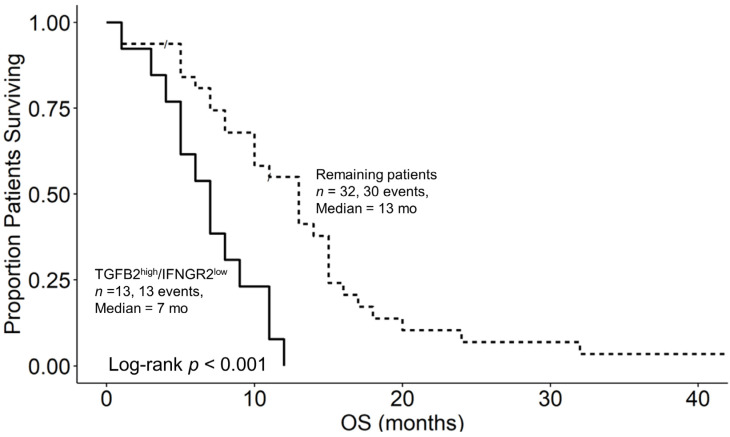
pbDMG patients with high levels of TGFB2 and low levels of IFNGR2 mRNA expression exhibited significantly shorter OS times than the remaining patients. We downloaded clinical metadata and RNA-sequencing-based mRNA expression data for 45 patients diagnosed with pbDMG (https://pedcbioportal.kidsfirstdrc.org/study/summary?id=openpbta%2Cpbta_all (accessed on 16 May 2023)). The RSEM-determined TPM metric was used to calculate the percentiles of TGFB2 and IFNGR2 expressions in 45 pbDMG patients. Two patient groups were then formed based on their expression levels of TGFB2 and IFNGR2: high expression of TGFB2 mRNA and low expression of IFNGR2 (TGFB2^high^/IFNGR2^low^; higher than or equal to that of TGFB2 in the 50th percentile and lower than that of IFNGR2 in the 50th percentile (*n* = 13)); and the remaining patients (*n* = 32; pooled patients from the TGFB2^low^/IFNGR2^high^, TGFB2^low^/IFNGR2^low^, and TGFB2^high^/IFNGR2^high^ groups). OS curves were then compared between these groups to assess the survival impact of the combination of the TGFB2 and IFNGR2 levels. In the TGFB2^high^/IFNGR2^low^ subset of patients, 8 were classified as DMG/H3K27M, 1 as DMG/H3K27M/TP53, 1 as HGG/H3 wildtype gene/IDH wildtype gene/TP53 mutation, 2 as HGG/to be classified, and 1 as NA according to the glioma grade and mutational status. In the remaining subset of patients, 15 were classified as DMG/H3K27M, 7 as DMG/H3K27M/TP53, 2 as HGG/H3 wildtype/IDH wildtype genes, and 8 as HGG/to be classified according to the glioma grade and mutational status. The mean (±SEM) and median (range) log2-TPM mRNA expression values of IFNGR2 in the TGFB2^high^/IFNGR2^low^ subset of patients were 4.7 ± 0.1 and 4.9 (3.7–5.4), respectively. For the remaining subset, the corresponding values were 5.6 ± 0.1 and 5.7 (3.2–6.9). The mean (±SEM) and median (range) log2-TPM mRNA expression values of TGFB2 in the TGFB2^high^/IFNGR2^low^ subset of patients were 5.3 ± 0.3 and 5.2 (4.1–7), respectively. For the remaining subset, the corresponding values were 3.5 ± 0.3 and 3.3 (0.6–5.7). In the group of 13 patients with TGFB2^high^/IFNGR2^low^, the median overall survival time was 7 months (95% CI = 5–NA months; 13 events). In contrast, the median overall survival time for the 32 patients in the remaining group was 13 months. The 95% CI was between 10 and 15, with 30 events. A statistically significant difference in the survival outcome was observed between the two groups, with a log-rank chi-square value of 13.5 and a *p*-value of 2.3 × 10^−4^.

**Figure 4 biomedicines-12-00191-f004:**
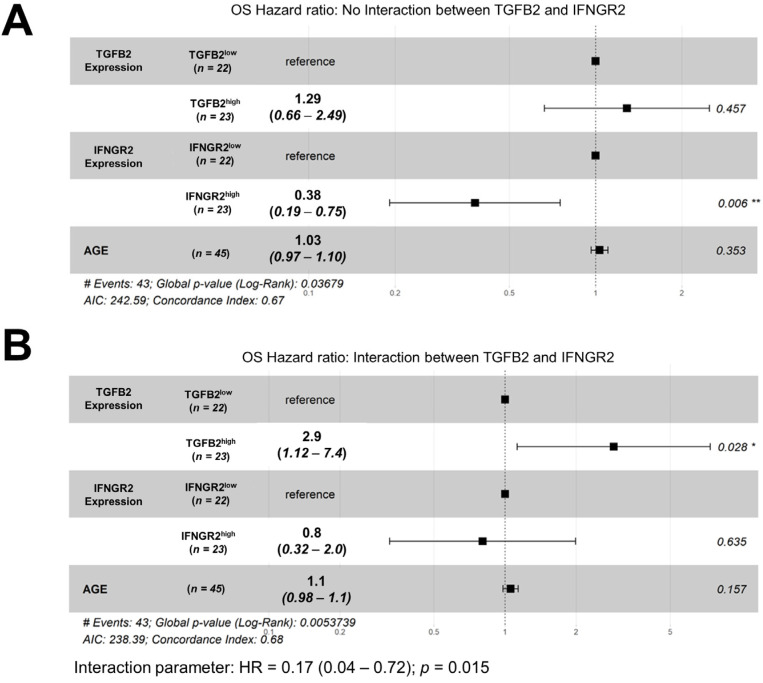
pbDMG patients with high levels of TGFB2 mRNA expression exhibited significantly increased hazard ratios in a multivariate Cox proportional hazards model considering age and the interaction between TGFB2 and IFNGR2. Multivariate analyses of the potential effects of TGFB2 and IFNGR2 levels on OS were determined using the multivariate Cox proportional hazards model to adjust for age by comparing models without (**A**) and with (**B**) TGFB2 and IFNGR2 interactions. Briefly, both models included (i) the mRNA expression level for TGFB2 as a categorical variable comparing high versus low TGFB2 mRNA expression levels (50% cutoff for the range of TPM values); (ii) the mRNA expression level for IFNGR2 as a categorical variable comparing high versus low IFNGR2 mRNA expression levels (50% cutoff for the range of TPM values); and (iii) age as a linear covariate. Forest plots were utilized to visualize the hazard ratios for Cox proportional hazards models for OS outcomes. We investigated the impact of including an interaction term as the fourth parameter in the Cox proportional hazards model (TGFB2 x IFNGR2) to compare the independent effects of TGFB2 and IFNGR2 in models with and without the interaction term. (**A**) The results of the model without an interaction term for the hazard ratios (HRs) showed that there was no significant increase in HR for the TGFB2^high^ group of patients (HR (95% CI range) = 1.29 (0.66–2.49); *p* = 0.457). However, there was a significant decrease in HR for the IFNGR2^high^ group of patients (HR (95% CI range) = 0.38 (0.19–0.75); *p* = 0.006). Additionally, there was no significant increase in HR for age as a linear covariate (HR (95% CI range) = 1.03 (0.97–1.1); *p* = 0.353). (**B**) The model that examined the interaction between IFNGR2 and TGFB2 uncovered a significant increase in the hazard ratio for patients in the TGFB2^high^ group (HR (95% CI range) = 2.88 (1.12–7.39); *p* = 0.028). However, there was no significant decrease in HR for patients in the IFNGR2^high^ group (HR (95% CI range) = 0.8 (0.32–1.99); *p* = 0.635) or for age as a linear covariate (HR (95% CI range) = 1.06 (0.98–1.14); *p* = 0.157). The results also showed the significant effect of the interaction term (HR (95% CI range) = 0.17 (0.04–0.72); *p* = 0.015). Overall, it was observed that the reduction in HR for the IFNGR2^high^ group of patients was independent of TGFB2 levels, and the inclusion of the interaction term in the model showed an increase in HR in the TGFB2^high^ group of patients and was independent of IFNGR2 levels and the interaction between TGFB2 and IFNGR2. The effect of IFNGR2 is captured in the interaction term. * denotes *p* < 0.05, ** denotes *p* < 0.01.

**Figure 5 biomedicines-12-00191-f005:**
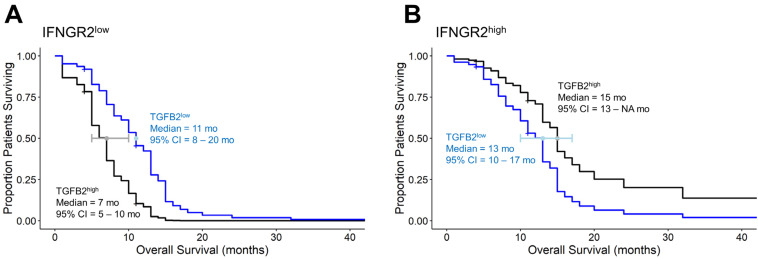
pbDMG patients with high levels of TGFB2 mRNA expression exhibited shorter OS times in the context of the low expression of IFNGR2 mRNA levels in the application of a multivariate Cox proportional hazards model considering age and the interaction between TGFB2 and IFNGR2 levels. The predicted survival proportion was calculated from the parameters in the Cox proportional hazards regression model, which included the interaction term for combinations of TGFB2 high and low mRNA expression groups in the context of IFNGR2^low^ (**A**) and IFNGR2^high^ (**B**) mRNA expression groups. The predicted survival curves plot the shift in the baseline OS curve for the 45 pbDMG patients by comparing the median OS times for TGFB2^high^ versus TGFB2^low^ groups of patients in patients who expressed either high or low levels of IFNGR2. (**A**) In the context of the low expression of IFNGR2, low levels of TGFB2 resulted in more favorable OS times, whereby the median OS time of 11 months for the TGFB2^low^ group of patients was greater than that of the upper 95% confidence limit for the TGFB2^high^ group of patients (upper 95% confidence interval = 10 months). (**B**) At higher levels of IFNGR2 expression, the TGFB2^high^ group of patients exhibited a median OS time of 15 months, which was within the 95% confidence interval for the TGFB2^low^ group of patients (median = 13 months; 95% CI = 10–17 months).

**Figure 6 biomedicines-12-00191-f006:**
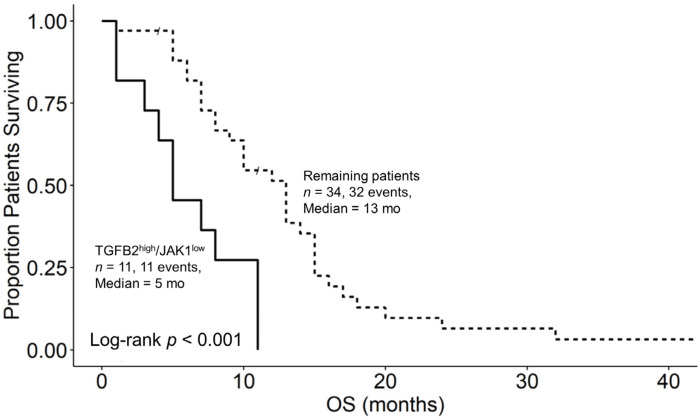
pbDMG patients with high levels of TGFB2 and low levels of JAK1 mRNA expression exhibited significantly shorter OS times than the remaining patients. We downloaded clinical metadata and RNA-sequencing-based mRNA expression data for 45 patients diagnosed with pbDMG (https://pedcbioportal.kidsfirstdrc.org/study/summary?id=openpbta%2Cpbta_all (accessed on 16 May 2023)). The RSEM-determined TPM metric was used to calculate the percentiles of TGFB2 and JAK1 expressions in the 45 pbDMG patients. Two patient groups were then formed based on their expression levels of TGFB2 and JAK1: high expression of TGFB2 mRNA and low expression of JAK1 (TGFB2^high^/JAK1^low^; higher than or equal to that of TGFB2 in the 50th percentile and lower than that of JAK1 in the 50th percentile (*n* = 11)); and the remaining patients (*n* = 34; pooled patients from the TGFB2^low^/JAK1^high^, TGFB2^low^/JAK1^low^, and TGFB2^high^/JAK1^high^ groups). In the TGFB2^high^/JAK1^low^ and remaining subsets of patients, 8 and 15 were respectively classified as DMG/H3K27M; 1 and 7 were respectively classified as DMG/H3K27M/TP53; and 2 and 8 were respectively classified as HGG/to be classified. Additionally, in the remaining group of patients, 2 were classified as possessing HGG/H3 wildtype/IDH wildtype genes, 1 as possessing HGG/H3 wildtype/IDH wildtype/TP53 mutation, and 1 was classified as NA according to the glioma grade and mutational status. The mean (±SEM) and median (range) log2-TPM mRNA expression values for JAK1 in the TGFB2^high^/JAK1^low^ subset of patients were 5.3 ± 0.1 and 5.4 (4.7–5.5), respectively. For the remaining subset, the mean (±SEM) and median (range) were 5.6 ± 0.1 and 5.7 (3.8–6.5), respectively. The mean (±SEM) and median (range) log2-TPM mRNA expression values for TGFB2 in the TGFB2^high^/JAK1^low^ subset of patients were 5.1 ± 0.2 and 5.2 (4.2–6.7), respectively. For the remaining subset, the mean (±SEM) and median (range) were 3.6 ± 0.3 and 3.4 (0.6–7), respectively. OS curves were then compared between these groups to assess the survival impact of the combination of the TGFB2 and JAK1 levels. The median OS time of 11 patients in the TGFB2^high^/JAK1^low^ group was 5 months (95% CI: 4–NA; number of events = 11), while the median OS time of the 34 patients in the remaining group was 13 months (95% CI: 9–15; number of events = 32). This difference in survival outcome between the two groups was statistically significant (log-rank chi-square value = 13.5; *p*-value = 2.4 × 10^−4^).

**Figure 7 biomedicines-12-00191-f007:**
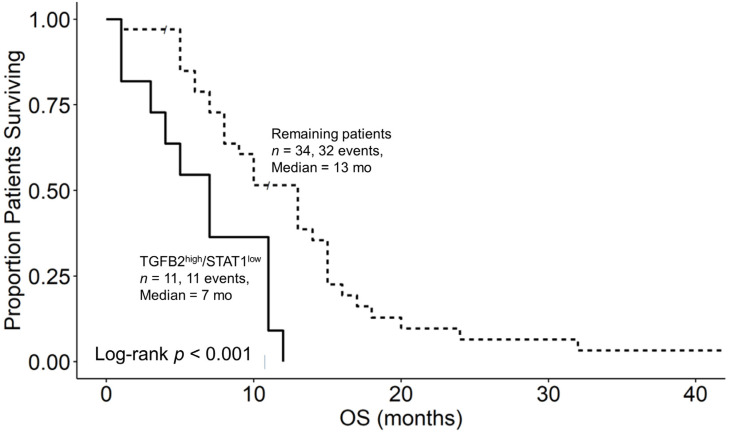
pbDMG patients with high levels of TGFB2 and low levels of STAT1 mRNA expression exhibited significantly shorter OS times than the remaining patients. We downloaded clinical metadata and RNA-sequencing-based mRNA expression data for the 45 patients diagnosed with pbDMG (https://pedcbioportal.kidsfirstdrc.org/study/summary?id=openpbta%2Cpbta_all (accessed on 16 May 2023)). The RSEM-determined TPM metric was used to calculate the percentiles of TGFB2 and STAT1 expressions in the 45 pbDMG patients. Two patient groups were then formed based on their expression levels of TGFB2 and STAT1: high expression of TGFB2 mRNA and low expression of STAT1 (TGFB2^high^/STAT1^low^; higher than or equal to that of TGFB2 in the 50th percentile and lower than that of STAT1 in the 50th percentile (*n* = 11)); and the remaining patients (*n* = 34; pooled patients from the TGFB2^low^/STAT1^high^, TGFB2^low^/STAT1^low^, and TGFB2^high^/STAT1^high^ groups). In the TGFB2^high^/STAT1^low^ subset of patients, 6 were classified as possessing DMG with the H3K27M mutation, 1 as possessing DMG with the H3K27M and TP53 mutations, 1 as possessing HGG harboring H3 wildtype and IDH wildtype genes, 2 as possessing HGG with no mutational classification, and 1 as NA. In the remaining subset of patients, 17 were classified as possessing DMG with the H3K27M mutation, 7 as possessing DMG with the H3K27M and TP53 mutations, 2 as possessing HGG harboring H3 wildtype and IDH wildtype genes, and 8 as possessing HGG with no mutational classification. The mean and median log2-TPM mRNA expression values for STAT1 in the TGFB2^high^/STAT1^low^ subset of patients were 4.3 ± 0.2 and 4.4 (3.4–5.1), respectively. These values were 5.4 ± 0.2 and 5.4 (3.1–8) for the remaining subset of patients. The mean and median log2-TPM mRNA expression values for the TGFB2^high^/STAT1^low^ subset of patients were 5 ± 0.2 and 5 (4.1–6.7), respectively. For the remaining subset of patients, these values were 3.7 ± 0.3 and 3.4 (0.6–7), respectively. OS curves were then compared between these groups to assess the survival impact of the combination of the TGFB2 and STAT1 levels. The median OS time for 11 patients in the TGFB2^high^/STAT1^low^ group was 7 months (95% CI: 4–NA; number of events = 11), while the median OS time for the 34 patients in the remaining group was 13 months (95% CI: 8–15; number of events = 32). The difference in the survival outcome between these groups was statistically significant (log-rank chi-square value = 10.3; *p*-value = 0.0014).

## Data Availability

All the data are presented in the manuscript and the Appendix A. The publicly available archived databases that were used to generate the data have been provided in the Materials and Methods section. Specifically, RNA-seq-determined expression values for normal pons specimens (rna_tissue_hpa.tsv.zip) were downloaded from https://www.proteinatlas.org/about/download (accessed on 17 December 2022);. Gene expression data files for CD14, CD163, and CD86; ITGAX; TGFB1, TGFB2, and TGFB3; JAK1; STAT1; and IFNGR2 were compiled from the RNA-seq expression data set repository stored at the cBioPortal for Cancer Genomics (https://pedcbioportal.kidsfirstdrc.org/study/summary?id=openpbta%2Cpbta_all (accessed on 16 May 2023)).

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
