# Peer review of "Transforming Growth Factor Beta 2 (TGFB2) and Interferon Gamma Receptor 2 (IFNGR2) mRNA Levels in the Brainstem Tumor Microenvironment (TME) Significantly Impact Overall Survival in Pediatric DMG Patients"

_biomedicines, 2024, doi:10.3390/biomedicines12010191_

Round 1

Reviewer 1 Report (Previous Reviewer 2)

Comments and Suggestions for Authors

Thank you, all of my previous comments have been addressed.

Author Response

“Thank you, all of my previous comments have been addressed.”

We thank the reviewer for the previous comments that enhanced the quality of this manuscript.

Reviewer 2 Report (Previous Reviewer 3)

Comments and Suggestions for Authors

although the topic is very promising, and some revisions have been made a bad taste remains:

- The authors now mention the diagnosisi of midline glioma by referring to WHO classification 2021, however, they still use roman number instead of arabic IV instead of 4 as propagated by WHO 2021

- The remaining tumors are not well characterized. At least the IDH-status should be given.

- The sentence ".. displayed features of both astrocytic and oligodendroglial cells” is non-sense. As soon as one detects oligodendroglial features, 1p/19q deletion has to be analysed.

Author Response

Response to Reviewer 2

“although the topic is very promising, and some revisions have been made a bad taste remains:

- The authors now mention the diagnosisi of midline glioma by referring to WHO classification 2021, however, they still use roman number instead of arabic IV instead of 4 as propagated by WHO 2021”

We have revised the classification as suggested, it can be seen highlighted in lines 147-149 and throught the manuscript.

“- The remaining tumors are not well characterized. At least the IDH-status should be given.

The database used in this manuscript only provided the IDH status for 3 of the HGG patients.  We have now included this information in the methods and figure legends for figure 3, 6 and 7 descriptions (highlighted).

“Thirteen tumors were designated as WHO Grade 3/4 high-grade gliomas (HGG), and one tumor had no designation (NA) [45]. Two HGG patients were reported to harbor both H3 and IDH wildtype genes, and one patient harbored H3 wildtype, IDH wildtype and a TP53 mutation.” (Lines 180-184)

“In the TGFB2high/IFNGR2low subset of patients, 8 were classified as DMG/H3K27M, 1 as DMG/H3K27M/TP53, 1 as HGG/H3 wildtype gene / IDH wildtype gene /TP53 mutation, 2 as HGG/To be classified, and 1 as NA according to glioma grade and mutational status. In the remaining subset of patients, 15 were classified as DMG/H3K27M, 7 as DMG/H3K27M/TP53, 2 as HGG/H3 wildtype/IDH wildtype genes, and 8 as HGG/To be classified according to glioma grade and mutational status.” (Lines 377-383)

“Additionally, in the remaining group of patients, 2 were classified as HGG/H3 wildtype/IDH wildtype genes, 1 as HGG/H3 wildtype/IDH wildtype/TP53 mutation,2 and 2 were respectively classified as HGG/H3 wildtype; 1 and 8 were respectively classified as HGG/To be classified; and 1 was classified as NA according to glioma grade and mutational status.” (Lines 483-486)

“1 as HGG harboring H3 wildtype and IDH wildtype genes, 2 as HGG with no mutation classification, and 1 as NA. In the remaining subset of patients, 17 w,ere classified as DMG with the H3K27M mutation, 7 as DMG with the H3K27M and TP53 mutations, 2 as HGG harboring H3 wildtype and IDH wildtype genes,” (Lines 504-508)

“- The sentence ".. displayed features of both astrocytic and oligodendroglial cells” is non-sense. As soon as one detects oligodendroglial features, 1p/19q deletion has to be analysed.”

We agree with the confusing nature of this statement that was compiled from the description of the patients reported in the archived database, and we have consequently removed the sentence.  No information was provided regarding the 1p/19q deletion status in the archived database.

Reviewer 3 Report (New Reviewer)

Comments and Suggestions for Authors

The authors discuss the role of TGFB2 in patients with DIPG.

Figure 1 should be at the end.

pbDMG patients with high levels of TGFB2 mRNA expression as compared to "remaining patients" seems to be a dubious compariosn.  Could you elaborate on the control and experimental arms of the study?

Comments on the Quality of English Language

n/a

Author Response

Response to Reviewer 3

“Figure 1 should be at the end.”

 We have now placed this figure at the end as suggested (revised Figure 8)

“pbDMG patients with high levels of TGFB2 mRNA expression as compared to "remaining patients" seems to be a dubious compariosn.  Could you elaborate on the control and experimental arms of the study?”

This was a retrospective analysis relating gene expression levels and survival outcomes. We ranked all patients according to TGFB2 levels and IFNGR2, or STAT1 or JAK1 levels at the 50th percentile cut-off.  We have now clarified in the figure legends the 2 arms of the comparisons.  There was no control or experimental arms.  The one arm of the comparison consisted of patients that expressed high levels of TGFB2 (greater than or equal to 50th percentile) AND low levels of IFNGR2, or STAT1 or JAK1, while the other arm was comprised of patients with 3 combinations of high or low expression levels of TGFB2 and high or low expression levels of  IFNGR2, or STAT1 or JAK1.

The updated figure legends now read:

“Two patient groups were then formed based on their expression levels of TGFB2 and IFNGR2: high expression of TGFB2 mRNA and low expression of IFNGR2 (TGFB2high/IFNGR2low; higher than or equal to the 50th percentile of both TGFB2 and lower than 50th percentile for IFNGR2 (N=13)); and remaining patients (N=32; pooled patients from the TGFB2low/IFNGR2high , TGFB2low/IFNGR2low , and TGFB2high/IFNGR2high groups).” (Figure 3)

“Two patient groups were then formed based on their expression levels of TGFB2 and JAK1: high expression of TGFB2 mRNA and low expression of JAK1 (TGFB2high/ JAK1low; higher than or equal to the 50th percentile of both TGFB2 and lower than 50th percentile for JAK1 (N=11)); and remaining patients (N=34; pooled patients from the TGFB2low/JAK1high , TGFB2low/JAK1low , and TGFB2high/JAK1high groups).” (Figure 6)

“Two patient groups were then formed based on their expression levels of TGFB2 and STAT1: high expression of TGFB2 mRNA and low expression of STAT1 (TGFB2high/ STAT1low; higher than or equal to the 50th percentile of both TGFB2 and lower than 50th percentile for STAT1 (N=11)); and remaining patients (N=34; pooled patients from the TGFB2low/STAT1high , TGFB2low/STAT1low , and TGFB2high/STAT1high groups).” (Figure 7)

Round 2

Reviewer 3 Report (New Reviewer)

Comments and Suggestions for Authors

Thank you for clarifying 

Comments on the Quality of English Language

N/a

This manuscript is a resubmission of an earlier submission. The following is a list of the peer review reports and author responses from that submission.

Round 1

Reviewer 1 Report

Comments and Suggestions for Authors

Comments on the Quality of English Language

Author Response

Response to Reviewer 1: 

We thank the reviewer for the criticisms and review comments.  Our responses to the specific review comments are provided point by point herein below:

“This is a potentially interesting study as it focuses attention on a very aggressive pediatric cancer with a very dismal prognosis, however, I found it very difficult to read and understand. The introduction, as well as the discussion, are very long and dispersive, leading to the loss of the manuscript focus. I tried reading the M&Ms in more detail but found them very confusing. There are no paragraphs, they are pooled together and not organized in a comprehensible way”.

We have reorganized both the introduction and discussion sections to focus more on the role of immune cells in the tumors based on the analysis of the mRNA  expression and prognostic indications for antigen-presenting cell markers CD14, CD163, CD86 and ITGAX/CDllc (Figures S1 and S2). The number of citations was updated to include 25 additional articles to better represent the findings of our investigations.

The revised introduction is outlined as follows:

-The prevalence and definition of Pediatric diffuse intrinsic pontine glioma (pDIPG).

-Summary of clinical trials that target treatment of pDIPG employing a diverse range of immune-based therapies.

-We then developed the rationale for targeting the abrogation of Transforming growth factor beta 2(TGFB2) ligand produced by tumor-promoting immune cells in conjunction with activation of the anti-tumor Interferon-gamma receptor pathway as a treatment option for pDIPG.

“We conducted this research to investigate the impact of the Transforming growth factor beta (TGFB) ligands active in the tumor microenvironment (TME) to adversely affect clinical outcomes in pDIPG patients, thereby expanding potential list of targets for treating these patients. TGFB ligands play a role in generating tumors by promoting immune suppression, immune evasion, and tumor progression. The TGFB signaling pathway is known for its immunosuppressive effects, which involve the inhibition of CD8-antigen-positive cytotoxic T cells and natural killer cells, as well as the activation of regulatory T cells [26] [27]  [28]  [29]  [30] [31,32] [33]. Our working hypothesis proposes that monocyte-derived M2 macrophages produce angiogenic and anti-inflammatory factors, such as IL-10 and TGFB ligands to promote regulatory T-cell infiltration  [34–36] that can be counterbalanced by activation of anti-tumor CD86 expressing M1 macrophages by interferon-gamma (IFN-γ)”.

We summarized the current understanding of the role of M2 macrophages identified by the expression of key cell surface markers and the involvement of these macrophages in various processes, including inflammation, tissue repair, and tumor growth. Two subtypes of M2-like macrophages produce TGFB ligands that maintain the tumor microenvironment in an immunosuppressive state. We state that,

“Therefore, targeting the TGFB ligands through inhibition or RNA interference has been identified as a potential therapeutic strategy to promote an anti-glioma immune response [46] [33]. This has been demonstrated in studies utilizing convection-enhanced delivery (CED) of OT101, a TGFB2-specific synthetic phosphorothioate antisense oligodeoxynucleotide (S-ODN), to treat recurrent/refractory WHO Grade 3 anaplastic astrocytoma (R/R AA) patients.”

Our working hypothesis for the treatment of cold tumors such as in pDIPG patients will require polarization of M0 to M1-like macrophages via stimulation with IFN-γ as stated in the updated introduction:

“We suggest that inhibiting TGFB2 production and release from immunosuppression in pDIPG tumors with concurrent activation of M0-derived M1 macrophages, characterized by the expression of CD80/CD86 and CCR7, via interferon-gamma (IFN-γ) may promote proinflammatory cytokine synthesis, enhanced phagocytosis, and increased tumor antigen-presenting capacity to improve the anti-tumor response to achieve better clinical outcomes [49].”

The methods and materials section has been revised with subsection headings to explain the processes of the bioinformatic calculations more clearly:

“2.1 Comparing mRNA expression levels in pDIPG tumors with normal brainstem/pons tissue.

2.2 pDIPG patient characteristics and stratification relative to mRNA expression levels.

2.3 OS outcomes of pDIPG patients stratified relative to TGFB2 and IFNGR2/JAK1/STAT1 mRNA expression levels.

2.4 Multivariate analysis of OS outcomes for pDIPG patients stratified relative to TGFB2 and IFNGR2 mRNA expression levels controlling for age, and interaction of TGFB2 and IFNGR2.

Similarly, the results section has been revised with subsection headings to explain the processes of the bioinformatic results more clearly:

“3.1 Downregulation of markers for anti-tumor antigen-presenting cells in pDIPG tumors.

3.2. Amplified expression of TGFB2 compared to TGFB1 and TGFB3 mRNA in pDIPG patients compared to normal pons tissue.

3.3. Amplified expression of IFNGR2 mRNA in pDIPG patients compared to normal pons tissue.

3.4. Amplified TGFB2 in combination with reduced IFNGR2 expression in pDIPG patients is associated with worse OS outcomes.

3.5. Amplified TGFB2 is an independent negative prognostic indicator for OS controlling for age and IFNGR2 levels.

3.6. Augmented TGFB2 mRNA levels and reduced levels of signaling molecules downstream of Interferon receptor activation are significant negative prognostic indicators for OS in pDIPG patients.”

The discussion section has been revised to include the role of immune cells based on cell surface maker expressions:

We investigated the mRNA expression profile of for markers of antigen-presenting cells in tumors isolated from pDIPG patients. In this analysis, the expression CD14, CD163, and ITGAX mRNA exhibited significant decreases of 1.64-fold (P=0.037), 1.75-fold (P=0.019), and 3.33-fold (P<0.0001), respectively in pDIPG tumors relative to normal brainstem/pons samples. CD86 mRNA expression in pDIPG patients showed a non-significant 1.42 decrease compared to normal brainstem/pons tissue (P=0.14) (Figure S1). In these pDIPG patients, high levels of TGFB2 expression in combination with low levels of CD14 (Figure S2A), CD163 (Figure S2B), and CD86 (Figure S2C) exhibited significantly worse OS outcomes than the remaining patients suggesting that abrogating TGFB2 and increasing the infiltration and/or function of antigen-presenting cells has the potential to markedly improve the prognosis of pDIPG patients.

The discussion expands on our present studies that provided further evidence that TGFB2 may be a key player in the development and progression of pDIPG with an updated set of patients, and that targeting TGFB2 and APCs may be a promising strategy for improving OS outcomes in these patients as stated in the revised discussion:

“The immune cold state of the pDIPG tumor microenvironment makes it challenging for treating these aggressive tumors [1]. Our study characterizes the prognostic impact of TGFB2 and IFNGR2 in brainstem tumors that provide potential insights into the prognostic role of these molecules in pDIPG patients. Several immune cells express receptors for interferon [56] and interferon-gamma (IFN-γ) is crucial in promoting a proinflammatory tumor environment and enhancing tumor immunogenicity by inducing M1 macrophages. M1 macrophages secrete IFN-γ, which helps in creating a proinflammatory microenvironment and promoting the development of T-cells responses [49]. Macrophages cannot become tumoricidal with IFN-γ alone because it requires combinations with toll-like receptor (TLR) agonists to induce macrophage tumoricidal activity. For the optimal development of antitumor M1 macrophages, two signals from the microenvironment, namely IFN-γ and TLR agonists, are thought to be necessary [73]. These present studies detail the interaction between TGFB2 and IFNGR2 in the tumor microenvironment that results in a prognostic impact on OS outcomes in pDIPG patients.”

We revised our conclusions based on the updated analysis to state,

“The expression of APC markers CD14, CD163, and ITGAX/CD11c mRNA exhibited significant decreases in pDIPG tumors relative to normal brainstem/pons samples.  pDIPG samples with high levels of TGFB2 mRNA in combination with low levels of APC markers, reflecting the cold immune state of pDIPG tumors, exhibited significantly worse overall survival outcome at low expression levels of CD14, CD163, and CD86. IFNGR2 and TGFB2 expression levels were significantly higher in pDIPG tumors than in normal brainstem/pons samples and high TGFB2 levels were linked to poor overall survival (OS) in pDIPG patients, regardless of IFNGR2 levels or patient age, and abrogating TGFB2 mRNA expression in the immunologically cold tumor microenvironment may help treat pDIPG patients. Additionally, pDIPG patients with low levels of JAK1 or STAT1 mRNA expression in combination with high levels of TGFB2 exhibited poor OS outcomes, suggesting that stimulating and activating JAK1 and STAT1 in anti-tumor APC cells using IFN-γ in the brainstem tumor microenvironment could enhance the effect of TGFB2 blockade.”

In response to reviewer 1 comment,

“FIG1 shows the graph of TGFB2, B2 and B3 mRNA expression with the related table (Fig1B) in DIPG
vs normal tissue, however the results are already published (PMID: 36980562). Although this article
is cited in the introductory section (Ref3), I suggest removing the already published results from the
panel of Fig1 and leaving only the unpublished ones”.

We have removed the TGFB1/2/3 barcharts and produced an additional supplementary Figure S3.  These findings were very similar to the previously published paper on DIPG from our group, but we used an updated dataset that included additional patients, so we still wanted to include the figure and results description in this paper.

Reviewer 2 Report

Comments and Suggestions for Authors

Authors analyze TGFB2 signaling in DIPG patients reusing open data.

The manuscript appears to be a draft construct and requires some refinements in content and formatting.

The introduction could be rewritten as the content seems to be a mere list of sometimes unconnected examples.

Line 75 please clarify the comparison, since the first group is part of DIPG patients.

Line 85-88 seem out of place/context & reference is mising.

Pleae provide all references from initial studies to used data and further adhere to citing rules of web resources.

Methods could always be detailed instead of referring to some old reference (avoiding as previously designed)!

Why are numbers of remaining patinents not the same for Figure 3 and Figure S1 if the number of the specified sample do match?

Continued success with your research!

Author Response

Response to Reviewer 2: 

We thank the reviewer for the criticisms and review comments.  Our responses to the specific review comments are provided point by point herein below:

Response to comments 1 and 2

“The manuscript appears to be a draft construct and requires some refinements in content and formatting.

The introduction could be rewritten as the content seems to be a mere list of sometimes unconnected examples”

We have extensively re-written the introduction discussion and conclusions based on an additional analysis investigating antigen-presenting cell marker cd14, CD163, CD86 and ITGAX/CD11c to better describe the immune response in the TME.

The revised introduction is outlined as follows:

-The prevalence and definition of Pediatric diffuse intrinsic pontine glioma (pDIPG).

-Summary of clinical trials that target treatment of pDIPG employing a diverse range of immune-based therapies.

-We then developed the rationale for targeting the abrogation of Transforming growth factor beta 2(TGFB2) ligand produced by tumor-promoting immune cells in conjunction with activation of the anti-tumor Interferon-gamma receptor pathway as a treatment option for pDIPG.

“We conducted this research to investigate the impact of the Transforming growth factor beta (TGFB) ligands active in the tumor microenvironment (TME) to adversely affect clinical outcomes in pDIPG patients, thereby expanding potential list of targets for treating these patients. TGFB ligands play a role in generating tumors by promoting immune suppression, immune evasion, and tumor progression. The TGFB signaling pathway is known for its immunosuppressive effects, which involve the inhibition of CD8-antigen-positive cytotoxic T cells and natural killer cells, as well as the activation of regulatory T cells [26] [27]  [28]  [29]  [30] [31,32] [33]. Our working hypothesis proposes that monocyte-derived M2 macrophages produce angiogenic and anti-inflammatory factors, such as IL-10 and TGFB ligands to promote regulatory T-cell infiltration  [34–36] that can be counterbalanced by activation of anti-tumor CD86 expressing M1 macrophages by interferon-gamma (IFN-γ)”.

We summarized the current understanding of the role of M2 macrophages identified by the expression of key cell surface markers and the involvement of these macrophages in various processes, including inflammation, tissue repair, and tumor growth. Two subtypes of M2-like macrophages produce TGFB ligands that maintain the tumor microenvironment in an immunosuppressive state. We state that,

“Therefore, targeting the TGFB ligands through inhibition or RNA interference has been identified as a potential therapeutic strategy to promote an anti-glioma immune response [46] [33]. This has been demonstrated in studies utilizing convection-enhanced delivery (CED) of OT101, a TGFB2-specific synthetic phosphorothioate antisense oligodeoxynucleotide (S-ODN), to treat recurrent/refractory WHO Grade 3 anaplastic astrocytoma (R/R AA) patients.”

Our working hypothesis for the treatment of cold tumors such as in pDIPG patients will require polarization of M0 to M1-like macrophages via stimulation with IFN-γ as stated in the updated introduction:

“We suggest that inhibiting TGFB2 production and release from immunosuppression in pDIPG tumors with concurrent activation of M0-derived M1 macrophages, characterized by the expression of CD80/CD86 and CCR7, via interferon-gamma (IFN-γ) may promote proinflammatory cytokine synthesis, enhanced phagocytosis, and increased tumor antigen-presenting capacity to improve the anti-tumor response to achieve better clinical outcomes [49].”

The methods and materials section has been revised with subsection headings to explain the processes of the bioinformatic calculations more clearly:

“2.1 Comparing mRNA expression levels in pDIPG tumors with normal brainstem/pons tissue.

2.2 pDIPG patient characteristics and stratification relative to mRNA expression levels.

2.3 OS outcomes of pDIPG patients stratified relative to TGFB2 and IFNGR2/JAK1/STAT1 mRNA expression levels.

2.4 Multivariate analysis of OS outcomes for pDIPG patients stratified relative to TGFB2 and IFNGR2 mRNA expression levels controlling for age, and interaction of TGFB2 and IFNGR2.

Similarly, the results section has been revised with subsection headings to explain the processes of the bioinformatic results more clearly:

“3.1 Downregulation of markers for anti-tumor antigen-presenting cells in pDIPG tumors.

3.2. Amplified expression of TGFB2 compared to TGFB1 and TGFB3 mRNA in pDIPG patients compared to normal pons tissue.

3.3. Amplified expression of IFNGR2 mRNA in pDIPG patients compared to normal pons tissue.

3.4. Amplified TGFB2 in combination with reduced IFNGR2 expression in pDIPG patients is associated with worse OS outcomes.

3.5. Amplified TGFB2 is an independent negative prognostic indicator for OS controlling for age and IFNGR2 levels.

3.6. Augmented TGFB2 mRNA levels and reduced levels of signaling molecules downstream of Interferon receptor activation are significant negative prognostic indicators for OS in pDIPG patients.”

The discussion section has been revised to include the role of immune cells based on cell surface maker expressions:

We investigated the mRNA expression profile of for markers of antigen-presenting cells in tumors isolated from pDIPG patients. In this analysis, the expression CD14, CD163, and ITGAX mRNA exhibited significant decreases of 1.64-fold (P=0.037), 1.75-fold (P=0.019), and 3.33-fold (P<0.0001), respectively in pDIPG tumors relative to normal brainstem/pons samples. CD86 mRNA expression in pDIPG patients showed a non-significant 1.42 decrease compared to normal brainstem/pons tissue (P=0.14) (Figure S1). In these pDIPG patients, high levels of TGFB2 expression in combination with low levels of CD14 (Figure S2A), CD163 (Figure S2B), and CD86 (Figure S2C) exhibited significantly worse OS outcomes than the remaining patients suggesting that abrogating TGFB2 and increasing the infiltration and/or function of antigen-presenting cells has the potential to markedly improve the prognosis of pDIPG patients.

The discussion expands on our present studies that provided further evidence that TGFB2 may be a key player in the development and progression of pDIPG with an updated set of patients, and that targeting TGFB2 and APCs may be a promising strategy for improving OS outcomes in these patients as stated in the revised discussion:

“The immune cold state of the pDIPG tumor microenvironment makes it challenging for treating these aggressive tumors [1]. Our study characterizes the prognostic impact of TGFB2 and IFNGR2 in brainstem tumors that provide potential insights into the prognostic role of these molecules in pDIPG patients. Several immune cells express receptors for interferon [56] and interferon-gamma (IFN-γ) is crucial in promoting a proinflammatory tumor environment and enhancing tumor immunogenicity by inducing M1 macrophages. M1 macrophages secrete IFN-γ, which helps in creating a proinflammatory microenvironment and promoting the development of T-cells responses [49]. Macrophages cannot become tumoricidal with IFN-γ alone because it requires combinations with toll-like receptor (TLR) agonists to induce macrophage tumoricidal activity. For the optimal development of antitumor M1 macrophages, two signals from the microenvironment, namely IFN-γ and TLR agonists, are thought to be necessary [73]. These present studies detail the interaction between TGFB2 and IFNGR2 in the tumor microenvironment that results in a prognostic impact on OS outcomes in pDIPG patients.”

We revised our conclusions based on the updated analysis to state,

“The expression of APC markers CD14, CD163, and ITGAX/CD11c mRNA exhibited significant decreases in pDIPG tumors relative to normal brainstem/pons samples.  pDIPG samples with high levels of TGFB2 mRNA in combination with low levels of APC markers, reflecting the cold immune state of pDIPG tumors, exhibited significantly worse overall survival outcome at low expression levels of CD14, CD163, and CD86. IFNGR2 and TGFB2 expression levels were significantly higher in pDIPG tumors than in normal brainstem/pons samples and high TGFB2 levels were linked to poor overall survival (OS) in pDIPG patients, regardless of IFNGR2 levels or patient age, and abrogating TGFB2 mRNA expression in the immunologically cold tumor microenvironment may help treat pDIPG patients. Additionally, pDIPG patients with low levels of JAK1 or STAT1 mRNA expression in combination with high levels of TGFB2 exhibited poor OS outcomes, suggesting that stimulating and activating JAK1 and STAT1 in anti-tumor APC cells using IFN-γ in the brainstem tumor microenvironment could enhance the effect of TGFB2 blockade.”

Reviewer 3 Report

Comments and Suggestions for Authors

Without any doubt one can say that the bioinformatical evaluation f the date are adequately done.

The points which have to be raised include:

- How was the tissue of brainstem gliomas obtained originally?

- How did the authors define tumor microenvironment from tumor cells?

-What were the WHO grades of the brainstem tumor?

- Was the mutational status of H3K27M determined?

- Are tissue sections available to show the described expression patterns using antibodies and immunohistochemistry.

- How were pedriatric DIPG patients defined in contrast to DIPG patients (line 75 and 76)

- The authors include Suppl Table S1 but fail to provide more information what was really done with these samples.

- Minor changes in the English style (line 105) ..calls can be bacause..)

Comments on the Quality of English Language

-Minor changes in the english style (line 105)

Author Response

Response to comment  3

Line 75 please clarify the comparison, since the first group is part of DIPG patients.

We thank the reviewer for alerting us to this ambiguity.  We have modified the language as follows:

“High levels of TGFB2 in pDIPG patients displayed significantly earlier progression-free survival and overall survival times compared to remaining pDIPG patients with low levels of TGFB2”

Response to comment  4 and 5

“Line 85-88 seem out of place/context & reference is mising.”

We have removed these lines from the revised manuscript.

“Pleae provide all references from initial studies to used data and further adhere to citing rules of web resources.

Methods could always be detailed instead of referring to some old reference (avoiding as previously designed)!”

We have modified the methods section in response to this reviewers concern and now states,

Antigen-presenting cell markers: CD14, CD163, CD86, ITGAX/CD11c, transforming growth factor receptor ligands: TGFB1, TGFB2, TGFB3, and IFN-γ receptor and downstream signaling molecules: IFNGR2, JAK1, and STAT1, mRNA transcript expression levels from RNA-seq experiments for brain tissue (rna_tissue_hpa.tsv.zip) were acquired from https://www.proteinatlas.org/ and accessed on 19th December 2022. We also downloaded mRNA expression levels of these genes from cBioPortal for Cancer Genomics to compare with the expression in pDIPG tumors using the reported RNAseq TPM values as detailed in [3]. Briefly, data arrays for the mRNA expression values for each gene were normalized to “transcripts per million” (“TPM”) values calculated using RSEM [60] software (https://pedcbioportal.kidsfirstdrc.org/study/summary?id=openpbta%2Cpbta_all (accessed on 4 August 2023)) [61,62] [63] [3].

We utilized TPM expression values from the 29 pons regions of the brain, filtering annotations under “Tissue Group” in the description file to compare with those of 45 pDIPG patients, by applying a two-way ANOVA model to identify differentially expressed genes. The log2 transformed TPM values for Gene (Model 1: TGFB1, TGFB2, TGFB3, JAK1, STAT1, and IFNGR2; Model 2: CD14, CD163, CD86, ITGAX/CD11c) and Tissue (29 normal pons tissues, 45 brainstem/pons specimens from pDIPG patients for both Models 1 and 2) were included as fixed factors, along with one interaction term to investigate gene level effects for normal and pDIG tissues (Gene x Tissue).  For each gene, we conducted a comparison between normal pons and pDIPG samples, and then determined significance by adjusting the P-value using false discovery rate algorithm provided for in the R-package (FDR corrected for all pairs in Model 1 and blocked design at the gene level for Model 2) calculations performed in R using multcomp_1.4-17 and emmeans_1.7.0  packages ran in R version 4.1.2 (2021-11-01) with Rstudio front end (RStudio 2021.09.0+351 "Ghost Orchid" Release)).  Bar chart graphics were constructed using the ggplot2_3.3.5 R package.

Response to comment  6

“Why are numbers of remaining patients not the same for Figure 3 and Figure S1 if the number of the specified sample do match?”

The dataset downloaded from the cBioportal indicated that 6 patients had no recordings of expression level measurements for TGFB1 (Figure S4) and TGFB3 (Figure S5), so that 39 patients were evaluated for the OS curves.

Round 2

Reviewer 2 Report

Comments and Suggestions for Authors

Thank you for the revised manuscript, though not all of my points have been adressed and some new issues have come up!

The introduction is still hard to follow, now due to other reasons:
A list of clinical trial indicators in the introduction section seems not to be best readable. This list could be presented as table, and in the introduction the content could be elaborated in some more detail if relevant.
It would be also interesting to summarize described elements of TGFb signaling in a Figure (possibly also as comparison of known elements to hypothesized elements).
Reference is missing in line 298.

Please cite specific/individual datasets used in the materials section. You could also describe the search keys used next to the access date.

There is still a discrepency of remaining patient numbers in Figures S4/S5, compared to e.g. Figure 3. The overall number would not be the same?

Author Response

Response to reviewer 2.

Comment 1

“The introduction is still hard to follow”

In response to this comment we have now added subsection headings in the introduction to guide the reader to the flow of the argument:

“1.1 Classification of Diffuse intrinsic pontine glioma (DIPG).”

“1.2 Clinical trials targeting Diffuse Intrinsic Pontine Gliomas (DIPG).”

“1.3 The role of Transforming growth factor beta (TGFB) receptor ligands in forming an immunosuppresive tumor microenvironment (TME).”

“1.4 Activation of interferon gamma receptor to promote an anti-tumor TME.”

Comment 2

“A list of clinical trial indicators in the introduction section seems not to be best readable. This list could be presented as table, “

In response to this comment, we have now included a supplementary Table S1 that provides details of the NCT number, Titles, Conditions, Interventions, Age, and clinical trial phase for the treatments targeting pDIPG.  We feel that this represents a significant improvement to the introduction.

Comment 3

“and in the introduction the content could be elaborated in some more detail if relevant.”

We have taken this suggestion and made the section more readable and provided some additional details with references to the publications for the trials that have shown promising leads.

“An urgent unmet need exists for effective treatment strategies to improve the very poor prognosis of pDIPG. Researchers have focused on adoptive transfer cell therapy, vaccines against H3.3K27M peptides, oncolytic virus therapy, and immunotherapies as current strategies for treatment (Table S1) [1]. GD2 CART therapy has entered 3 phase 1 trials for DIPG patients (NCT04196413, NCT04099797, NCT05298995) which is a type of cancer treatment that uses T-cells modified with chimeric antigen receptors (CARTs) to target and attack cancer cells that express the GD2 antigen. Since GD2 is a disialoganglioside that is expressed on the surface of cancer cells, this target is amenable to adoptive transfer cell therapy [10,11].  Cluster of Differentiation 276 (CD276)/ B7 Homolog 3 (B7-H3) is another cell surface expressed protein targetable by CARTs that has prompted the use of B7-H3-Specific CAR T-Cell immunotherapy for investigating DIPG treatment via a phase 1 clinical trial (NCT04185038). In this phase I trial, children with recurrent or refractory CNS tumors and DIPG were given repeated locoregional B7-H3 CAR T cells. The results indicate that the first three evaluable patients with DIPG, including two who enrolled after progression, received 40 infusions with no dose-limiting toxicities [12].

H3.3K27M is a mutation in the histone 3 variant 3 involving a lysine (K) to methionine (M) mutation at position 27 of the H3.3 protein that is highly expressed in diffuse intrinsic pontine glioma DIPG [13–16]. Three clinical trials have targeted the H3.3K27M mutation using vaccines; NCT02960230, NCT04749641, and NCT04808245. Published observations from NCT02960230 [17] showed that the H3.3K27M-specific vaccine had no grade-4 treatment-related adverse events, and those who had a CD8+ immunological response to H3.3K27M showed longer overall survival compared to those who did not respond. Patients with an expansion of H3.3K27M-reactive CD8+ T cells had a median overall survival of 16.1 months, while non-responder counterparts had a median overall survival of 9.8 months (P = 0.05). Other trials using vaccines include NCT01058850 and NCT04943848. 

Oncolytic virus therapy is a potential treatment option for diffuse intrinsic pontine glioma (DIPG). Several studies have investigated the use of oncolytic viruses, such as Delta-24-RGD (DNX-2401; NCT03178032) [18]. Four other trials are underway using oncolytic viruses to treat DIPG: NCT03330197; NCT02444546; NCT04758533; and, NCT05096481. Initial findings from the trial NCT03330197 [19] reported potentially a promising immunotherapy candidate for DIPG involving loco-regional delivery of interleukin 12, which is administered using the proprietary transcriptional switch RheoSwitch Therapeutic System® (RTS®) delivered via a replication-incompetent adenovirus. The therapy suggested effective eliciting of a tumor-specific effector immune response. The first DIPG subject who underwent this treatment showed encouraging data on safety, tolerability, serum cytokines, and early signs consistent with a clinical response.

Immunotherapy studies include NCT01952769 employing anti-PD1 Antibody MDV9300 (a monoclonal antibody that binds to programmed death-1 (PD-1), Pidilizumab) in DIPG; NCT02359565 which is a study that utilized Pembrolizumab in treating DIPG; and NCT03690869 which is a study using REGN2810 (a monoclonal antibody cemiplimab) in pediatric patients with newly diagnosed or recurrent gliomas.”

Comment 4

“It would be also interesting to summarize described elements of TGFb signaling in a Figure (possibly also as comparison of known elements to hypothesized elements).”

We thank the reviewer for this very useful suggestion and we have now included Figure 8 in the paper showing activation of TGFB and interferon gamma receptor activation along with the prognostic significance for the association of TGFB2, IFNGR2, JAK1, and STAT1 with OS in pDIPG patients.

Comment 5

“Reference is missing in line 298.”

We thank the reviewer for noticing this omission and we have now included the reference.

“IFN-γ targetable cellular level responses that upon the release of immunosuppression by high TGFB2 levels can augment the antitumor response [50].”

Comment 6

“Please cite specific/individual datasets used in the materials section. You could also describe the search keys used next to the access date.”

In response to this comment, we have taken this suggestion and provided the search keys for the datasets.

“We compiled TPM expression values from only the pons regions of the brain by filtering “Tisssue Group” annotations in the accompanying description file ("rna_tissue_hpa_description.tsv.zip").  This data file included average levels of gene expression in 29 pons regions filtered using the key, “pons” that retrieved values from the following regions:  "anterior cochlear nucleus, ventral", "dorsal cochlear nucleus" , "dorsal tegmental nucleus",  "dorsolateral tegmental area", "Kolliker-Fuse nucleus",  "lateral lemniscal nuclei" , "lateral parabrachial nucleus",  "lateral vestibular nucleus", "locus coeruleus", "medial olivary nucleus", "medial parabrachial nucleus", "medial periolivary nuclei", "motor facial nucleus",  "motor trigeminal nucleus", "nuclei of the trapezoid body" ,  "paramedian reticular nucleus" , "pontine nuclei",  "pontine raphe nucleus","posteroventral cochlear nucleus",  "principal sensory trigeminal nucleus", "reticular pontine nucleus, caudal",  "reticular pontine nucleus, oral", "reticulotegmental nucleus", "spinal trigeminal nucleus, oral", "subcoeruleus area", "superior olive", "superior vestibular nucleus", "ventral periolivary nuclei", "ventrolateral tegmental area, A5 NE cell group".

We also downloaded mRNA expression levels of these genes from PedcBioPortal for Childhood Cancer Genomics to compare with the expression in pDIPG tumors using the reported RNAseq TPM values as detailed in [3]. Briefly, data arrays for the mRNA expression values for each gene were normalized to “transcripts per million” (“TPM”) for gene abundance values calculated using RSEM [57] software (dataset downloaded from the PedcBioportal, https://pedcbioportal.kidsfirstdrc.org/study/summary?id=openpbta%2Cpbta_all compiled using the Open Pediatric Brain Tumor Atlas (OpenPBTA) and Pediatric Brain Tumor Atlas (PBTA, Provisional) consortiums [58](accessed on 4 August 2023: keys for the search were “Brainstem glioma, Diffuse intrinsic pontine glioma, Diffuse midline glioma Grade IV, Diffuse Midline Glioma H2K27M WHO grade IV, diffuse midline glioma WHO grade IV H3K27M mutant, DMG H3 K27M-Mutant WHO Grade IV, diffuse midline high-grade glioma, diffuse hemispheric glioma H3 G34-mutant, WHO grade 4, and Infiltrating Dipg”.)). The PedcBioportal enables acquisition of CSV formatted files for the compiled clinical metadata and expression values for the filtered patient subsets for further analyses [59,60] [61] [3] [62].”

“Patient characteristics for these 45 pDIPG patients were compared with 171 pediatric High-Grade Gliomas and 404 Low-Grade Gliomas obtained from the pbta database (downloaded from the PedcBioportal, https://pedcbioportal.kidsfirstdrc.org/study/summary?id=openpbta%2Cpbta_all compiled using the Open Pediatric Brain Tumor Atlas (OpenPBTA) and Pediatric Brain Tumor Atlas (PBTA, Provisional) consortiums [58](accessed on 4 August 2023: keys for the High-grade glioma and Low-grade Gliomas search were: “CANCER_TYPE_DETAILED : Low-Grade Glioma, NOS or High-Grade Glioma, NOS RNA expression” with “ONCOTREE_CODE: dipg, hggnos and lggnos).”

Comment 7

“There is still a discrepency of remaining patient numbers in Figures S4/S5, compared to e.g. Figure 3. The overall number would not be the same?”

We have now clarified in the figure legends that mRNA expression levels were not reported for 6 pDIPG patients so only 39 patients could be evaluated for TGFB1 and TGFB3 comparisons.

For Figure S3: “TGFB1 (N=39; mRNA expression values were not recorded for 6 patients), TGFB2(N=45), TGFB3 (N=39; mRNA expression values were not recorded for 6 patients)”

For Figure S4 and S5: “The RSEM-determined TPM metric was used to calculate the percentiles of TGFB1 and IFNGR2 expression in 39 evaluable pediatric DIPG patients (mRNA expression values were not recorded for 6 patients).”

Reviewer 3 Report

Comments and Suggestions for Authors

The revised manuscript is still lacking information about:

- how was normal tissue from the pons/brainstem obtained

- histological diagnoses of pDIPG

Author Response

Response to Reviewer 3.

Comment 1

“how was normal tissue from the pons/brainstem obtained”

In response to this comment, we have now provided some details regarding the normal tissue acquisition for our analysis and provided source references for the dataset.

“Antigen-presenting cell markers: CD14, CD163, CD86, ITGAX/CD11c, transforming growth factor receptor ligands: TGFB1, TGFB2, TGFB3, and IFN-γ receptor and downstream signaling molecules: IFNGR2, JAK1, and STAT1, mRNA transcript expression levels from RNA-seq experiments for brain tissue (rna_tissue_hpa.tsv.zip) were acquired from the Human Protein Atlas version 23.0 (https://www.proteinatlas.org/ and accessed on 19th December 2022; search key was “pons”). Human tissues were anatomically dissected and analyzed using transcriptomics based on mRNA samples from normal tissues extracted from frozen tissue sections. Following sequencing, alignment and quantification of the extracted nuclear RNA, the genes were annotated using Ensembl version 109 [55,56].”

Comment 2

“ histological diagnoses of pDIPG”

The clinical data file suggested all 45 patients were characterized as having diffuse astrocytic and oligodendroglial tumors.

“The clinical metadata and RNA sequencing data for 45 patients diagnosed with Grade IV pediatric pDIPG, all of whom had diffuse astrocytic and oligodendroglial tumors. “

Round 3

Reviewer 2 Report

Comments and Suggestions for Authors

Readability of the text in the revised version has increased - except for strikedthrough passages which make it hard to distinguish between final sentences (some brackets may be missing e.g. line 97-99??263?).
Line 108+ still reads like a list, as NCT numbers would suffice to be mentioned in the table with a reference in the text, not to interfere with the reading flow.

Methods are now described in a reproducible manner. References to the original source studies of datasets are not mandatory but recommended.

Thank you also for the new Figure, it could be used already in the beginning, in the introduction, to give a first signalling overview and to highlight the hypothesis/outcome of the paper.

Author Response

“Readability of the text in the revised version has increased - except for strikedthrough passages which make it hard to distinguish between final sentences (some brackets may be missing e.g. line 97-99??263?).

Line 108+ still reads like a list, as NCT numbers would suffice to be mentioned in the table with a reference in the text, not to interfere with the reading flow.”

We have added the missing brackets, revised the clinical trials paragraph, and replaced the NCT numbers with reference to Table S1.

We added 3 additional references that present initial published results from the following trials:

“Several studies have investigated the use of oncolytic viruses (Table S1), such as Del-ta-24-RGD (DNX-2401), that, when combined with subsequent radiotherapy in pDIPG patients, the treatment demonstrated altered T-cell activity and tumor reduction in 9 out of 12 patients (Table S1) [18].”

“Immunotherapy in pDIPG that employed anti-PD1 antibody MDV9300/Pidilizumab (a monoclonal antibody that binds to programmed death-1 (PD-1) receptor) showed administration of a total of 30 cycles of Pidilizumab was well tolerated with transient fatigue as the main side effect, and 2 patients exhibited a reduction in tumor size [20].  Two other trials utilizing Pembrolizumab (anti-PD⁠-⁠1 antibody), and REGN2810 (a monoclonal an-ti-PD1 antibody, cemiplimab) in pediatric patients with newly diagnosed or recurrent gliomas have not reported any results to date (Table S1).”

“Immunotherapy in pDIPG that employed anti-PD1 antibody MDV9300/Pidilizumab (a monoclonal antibody that binds to programmed death-1 (PD-1) receptor) showed administration of a total of 30 cycles of Pidilizumab was well tolerated with transient fatigue as the main side effect, and 2 out of 6 patients exhibited a reduction in tumor size [20]. “

“Methods are now described in a reproducible manner. References to the original source studies of datasets are not mandatory but recommended.”

We thank the reviewer for recognizing the improvement in the methods section.

“Thank you also for the new Figure, it could be used already in the beginning, in the introduction, to give a first signaling overview and to highlight the hypothesis/outcome of the paper.”

We have now made the cartoon figure the first figure in the paper.

Reviewer 3 Report

Comments and Suggestions for Authors

The authors did not understand the meaning of the reviewer´s comments:

- what was the diagnosis and WHO grade of the analysed samples. It is not sufficient o answere that the various grades were retrieved in the data bases. Also the answer provided shows the lack ok understanding of brain tumors

Author Response

“- what was the diagnosis and WHO grade of the analysed samples. It is not sufficient o answere that the various grades were retrieved in the data bases. Also the answer provided shows the lack ok understanding of brain tumors.”

Our group is initiating a clinical trial involving DIPG patients, and complete analysis of patient characteristics using diagnostic radiological, immunohistochemical methods are being analyzed and will be more fully reported in subsequent publications.  We have now provided additional details regarding the diagnosis and grade designation of the pDIPG tumors for the secondary data analysis in this report.

“The clinical metadata and RNA sequencing data for 45 pDIPG patients were analyzed to stratify patients according to mRNA expression levels in this study. The diagnosis of pDIPG was primarily ascertained using radiological methods, which revealed borderless, diffuse expansile hyperintense lesions in the pons that extended into other areas of the brainstem for all 45 patients. Upon examining the clinical metadata file, it was reported that 31 out of 45 patients had Diffuse Midline Gliomas with H3K27M mutation (DMG/H3K27M), which designates these tumors as Grade IV according to the WHO 2021 classification scheme. Thirteen tumors were designated as WHO Grade III/IV high-grade gliomas (HGG), and one tumor had no designation (NA).  All these tumors from 45 patients displayed features of both astrocytic and oligodendroglial cells”